# Loss of hepatic aldolase B activates Akt and promotes hepatocellular carcinogenesis by destabilizing the Aldob/Akt/PP2A protein complex

Xuxiao He[1,2©], Min Li[1,2©], Hongming Yu[3], Guijun Liu[1,2], Ningning Wang[1,2], Chunzhao Yin[2,4], Qiaochu Tu[2,4], Goutham Narla[5], Yongzhen Tao[1]*, Shuqun Cheng[3]*, Huiyong Yin[1,2,4,6‡]*

**1** CAS Key Laboratory of Nutrition, Metabolism and Food Safety, Shanghai Institute of Nutrition and Health, Chinese Academy of Sciences (CAS), Shanghai, China, **2** University of Chinese Academy of Sciences, Chinese Academy of Sciences, Beijing, China, **3** The Eastern Hepatobiliary Surgery Hospital, Shanghai, China, **4** School of Life Science and Technology, ShanghaiTech University, Shanghai, China, **5** Division of Genetic Medicine, Department of International Medicine, University of Michigan, Ann Arbor, Michigan, United States of America, **6** Key Laboratory of Food Safety Risk Assessment, Ministry of Health, Beijing, China

© These authors contributed equally to this work.
‡ Lead contact.
* yztao01@sibs.ac.cn (YT); chengshuqun@aliyun.com (SC); hyyin@sibs.ac.cn (HY)

**Data Availability Statement:** All relevant data are within the paper and its Supporting Information files.

## Abstract

Loss of hepatic fructose-1, 6-bisphosphate aldolase B (Aldob) leads to a paradoxical up-regulation of glucose metabolism to favor hepatocellular carcinogenesis (HCC), but the upstream signaling events remain poorly defined. Akt is highly activated in HCC, and targeting Akt is being explored as a potential therapy for HCC. Herein, we demonstrate that Aldob suppresses Akt activity and tumor growth through a protein complex containing Aldob, Akt, and protein phosphatase 2A (PP2A), leading to inhibition of cell viability, cell cycle progression, glucose uptake, and metabolism. Interestingly, Aldob directly interacts with phosphorylated Akt (p-Akt) and promotes the recruitment of PP2A to dephosphorylate p-Akt, and this scaffolding effect of Aldob is independent of its enzymatic activity. Loss of Aldob or disruption of Aldob/Akt interaction in Aldob R304A mutant restores Akt activity and tumor-promoting effects. Consistently, Aldob and p-Akt expression are inversely correlated in human HCC tissues, and Aldob down-regulation coupled with p-Akt up-regulation predicts a poor prognosis for HCC. We have further discovered that Akt inhibition or a specific small-molecule activator of PP2A (SMAP) efficiently attenuates HCC tumorigenesis in xenograft mouse models. Our work reveals a novel nonenzymatic role of Aldob in negative regulation of Akt activation, suggesting that directly inhibiting Akt activity or through reactivating PP2A may be a potential therapeutic approach for HCC treatment.

**Funding:** This work was financially supported by the National Key R&D Program of China administered by Chinese Ministry of Science and Technology (MOST) (2018YFA0800300 for HY) and National Natural Science Foundation of China (91857112, 31671231, and 32030053 for HY). The funders had no role in study design, data collection and analysis, decision to publish, or preparation of the manuscript.

**Competing interests:** I have read the journal's policy and the authors of this manuscript have the following competing interests: The Icahn School of Medicine at Mount Sinai has filed patents covering composition of matter on the small molecules disclosed herein for the treatment of human cancer and other diseases (International Application Numbers: PCT/US15/19770, PCT/US15/19764; and US Patent: US 9,540,358 B2). Mount Sinai is actively seeking commercial partners for the further development of the technology. G.N. has a financial interest in the commercialization of the technology. The other authors have declared that no competing interests exist.

**Abbreviations:** AFP, a-fetoprotein; Aldob, aldolase B; AMPK, AMP-activated protein kinase; CAS, Chinese Academy Science; DEN, diethyl nitrosamine; DMA, Dimethylacetamide; DMEM, Dulbecco's Modified Eagle Medium; EGF, epidermal growth factor; ERK, extracellular signal-regulated kinase; esiRNA, endoribonuclease-prepared siRNA; FBP, fructose-1, 6-bisphosphate; FBS, fetal bovine serum; GAP, GTPase-activating protein; GCK, glucokinase; GSK-3, glycogen synthase kinase-3; GSK3ß, glycogen synthase kinase 3ß; GST, glutathione S-transferase; G6PD, glucose-6-phosphate dehydrogenase; HCC, hepatocellular carcinoma; HFI, hereditary fructose intolerance; HK1, hexokinase 1; HK2, hexokinase 2; HRP, horseradish peroxidase; H&E, hematoxylin and eosin; IP, immunoprecipitation; IHC, immunohistochemistry; KO, knockout; mTOR, mechanistic target of rapamycin; mTORC1, mechanistic target of rapamycin complex 1; mTORC2, mechanistic target of rapamycin complex 2; OA, okadaic acid; PAGE, polyacrylamide gel electrophoresis; p-Akt, phosphorylated Akt; PDK1, phosphoinositide-dependent kinase 1; PHLPP, PH domain and leucine-rich repeat protein phosphatase; PHs, primary hepatocytes; PI, propidium iodide; PIP3, phosphatidylinositol-3, 4, 5-trisphosphate; PI3K, phosphoinositide 3-kinase; PKCa, protein kinase Ca; PPP, pentose phosphate pathway; PP2A, protein phosphatase 2A; PTEN, phosphatase and tensin homolog; PVDF, polyvinylidene fluoride; RNAi, RNA-mediated

## Introduction

Hepatocellular carcinoma (HCC) is the most common primary liver malignancy and the fourth leading cause of cancer-related deaths worldwide [1]. Despite considerable improvement in the diagnosis and treatment for HCC, the clinical prognosis of HCC remains disappointing primarily as a result of an incomplete understanding of the molecular mechanisms underlying HCC progression and limited therapeutic options [2].

Abnormal activation of the PI3K/Akt signaling pathway is a hallmark of many human malignancies, including HCC [3]. In response to growth factors or cytokines, activated PI3K generates phosphatidylinositol-3, 4, 5-trisphosphate (PIP$_3$), which recruits Akt for phosphorylation at threonine 308 (T308) by phosphoinositide-dependent kinase 1 (PDK1) and serine 473 (S473) by mechanistic target of rapamycin complex 2 (mTORC2) [4]. Activated Akt triggers the subsequent cellular response through the phosphorylation of various downstream substrates or induction of multiple gene expressions. For instance, inhibitory phosphorylation of glycogen synthase kinase 3β (GSK3β) by active Akt leads to increased stability and accumulation of CyclinD1, thereby promoting cell cycle progression [5]. Besides, activated Akt enhances the expression and activity of numerous glycolytic enzymes, such as glucose transporter, hexokinase, and phosphofructokinase, resulting in the up-regulation of glucose uptake and glycolysis [6,7]. Importantly, a series of feedback controls counteract Akt activation to maintain the transient signal. In addition to signal termination by phosphatase and tensin homolog (PTEN), protein phosphatase 2A (PP2A), and PH domain and leucine-rich repeat protein phosphatase (PHLPP) function as Akt phosphatases directly dephosphorylating Akt [8,9]. Therefore, a delicate balance between protein kinase-catalyzed phosphorylation and protein phosphatase-mediated dephosphorylation is vital for Akt kinase activity in cellular homeostasis; dysregulation of this balance may lead to tumorigenesis. However, the upstream signaling networks involved in the regulation of Akt activity remain to be fully characterized.

Fructose-1, 6-bisphosphate aldolase catalyzes the cleavage of fructose-1, 6-bisphosphate (FBP) to glyceraldehyde-3-phosphate and dihydroxyacetone phosphate in glycolysis. Aldolase B (Aldob) is abundantly expressed in the liver, kidney, and small intestine, whereas aldolase A (Aldoa) and aldolase C (Aldoc) are the muscle isoform and central nervous system isoform, respectively [10]. In humans, gene dysfunction or deficiency of Aldob causes hereditary fructose intolerance (HFI), a recessively inherited disorder of fructose metabolism [11]. Aldolase has been implicated in diverse physiological and pathological processes [12]. A recent study has identified that aldolase acts as a sensor of FBP and glucose availability in the regulation of AMP-activated protein kinase (AMPK) [13]. Dissociation of aldolase from the actin cytoskeleton leads to increased aldolase activity and enhanced glycolytic flux, which is positively regulated by PI3K signaling [14]. Aldob has been documented to be down-regulated in HCC tissues in transcriptomic and proteomic studies, which is correlated with multiple malignant characteristics of HCC [15–17]. The mechanism underlying the paradoxical loss of Aldob and up-regulated glycolysis in HCC tumor cells has not been defined until our recent study identified a novel tumor-suppressive mechanism by which Aldob directly binds and inhibits the rate-limiting enzyme in pentose phosphate pathway (PPP), glucose-6-phosphate dehydrogenase (G6PD) [18]. Loss of Aldob releases G6PD activity and up-regulates PPP metabolism to bypass the disrupted glycolysis. However, the upstream signaling events leading to HCC progression and metabolic reprogramming due to the loss of Aldob remain unknown.

Here, we report that Aldob negatively regulates Akt activation, which is required for Aldob-induced suppression of cancer cell proliferation, glucose uptake, and metabolism for tumorigenesis. Aldob directly binds to p-Akt and potentiates PP2A interaction and dephosphorylation of p-Akt, resulting in the inhibition of Akt phosphorylation and downstream oncogenic

interference; SDS, sodium dodecyl sulfate; SIBS, Shanghai Institutes for Biological Sciences; siRNA, small interfering RNA; SMAP, small-molecule activator of PP2A; TCA cycle, tricarboxylic acid cycle; TMA, tissue microarray; TSC2, tuberous sclerosis complex 2; T308, threonine 308; S473, serine 473; WT, wild-type.

signaling. This novel protein interaction appears to be independent of Aldob enzymatic activity. Moreover, a novel small molecule activator of PP2A (SMAP) elicits antitumor efficacy comparable to Akt allosteric inhibitor (MK2206) in blocking the tumorigenic effects driven by Aldob deficiency in vitro and in vivo. Interestingly, we have discovered an inverse correlation between Aldob and p-Akt expression in HCC tumor tissues and that a combination of low Aldob and high p-Akt expression is associated with the worst prognosis for HCC patients. Collectively, our work highlights that targeting Akt is a viable approach to treat HCC and PP2A activation using small molecule modulators of this phosphatase has the potential as a new targeted approach for HCC treatment.

## Results

### Aldob expression is negatively correlated with Akt activation in human HCC and the expression of Aldob/p-Akt predicts overall survival of HCC patients

Our recent study has revealed a novel tumor suppressive role for the glycolytic enzyme Aldob in HCC through directly binding to G6PD and inhibiting its activity, acting as a metabolic switch in glucose metabolism and regulating the metabolic reprogramming [18]. Accumulating body of evidence has demonstrated that metabolic reprogramming in cancer cells to meet increased bioenergetic and biosynthetic requirements during tumorigenesis depends on multiple intracellular signaling pathways [19]. Among them, activated Akt signaling has a profound impact on metabolic reprogramming through up-regulating glycolytic enzymes and promoting aerobic glycolysis [20]. Therefore, we set out to examine the role of Akt signaling in HCC in the context of Aldob down-regulation and metabolic reprogramming. We first investigated the clinical relevance between Aldob expression and Akt activation in human HCC. As shown in Fig 1A, reduced Aldob expression and enhanced Akt phosphorylation levels at both the threonine 308 (pT308-Akt) and serine 473 (pS473-Akt) residues were observed in tumor tissues as compared to matched adjacent normal liver tissues. Next, we performed tissue microarray (TMA) analysis on paired clinical samples from HCC patients (Figs 1B and S1A and S1 Table, $n = 70$). Aldob expression was negatively correlated with Akt activation in human HCC tissues (Fig 1C, $R^2 = 0.146$, $p = 0.001$). We also found that low Aldob expression was significantly correlated with α-fetoprotein (AFP), albumin level, and tumor encapsulation (S1 Table). Kaplan–Meier survival analysis showed that the overall survival time for patients with low Aldob expression was notably shorter than those with high Aldob expression (Fig 1D, $p = 0.016$). Conversely, patients with high pT308-Akt level exhibited shorter overall survival compared with patients with low pT308-Akt level (Fig 1E, $p = 0.029$). In addition, an elevated level of p-Akt was significantly correlated with prothrombin time, number of tumors, and tumor encapsulation (S1 Table). More importantly, a low Aldob expression coupled with high pT308-Akt levels was associated with the worst overall prognosis (Fig 1F, $p = 0.001$). In conclusion, our study uncovers a novel inverse correlation between Aldob and p-Akt expression in human HCC and low Aldob expression with high p-Akt predicts the worst prognosis for HCC patients, suggesting an important role of Akt signaling in HCC in the context of Aldob downregulation.

### Aldob inhibits Akt kinase activity and downstream signaling in HCC

To investigate the relationship of Aldob down-regulation and activation of Akt signaling, we observed that, in liver-specific *ALDOB* knockout (KO) (ALDOB^f/f; *Alb*-Cre) mouse primary hepatocytes (PHs), Aldob deletion significantly augmented Akt activation compared to wild-

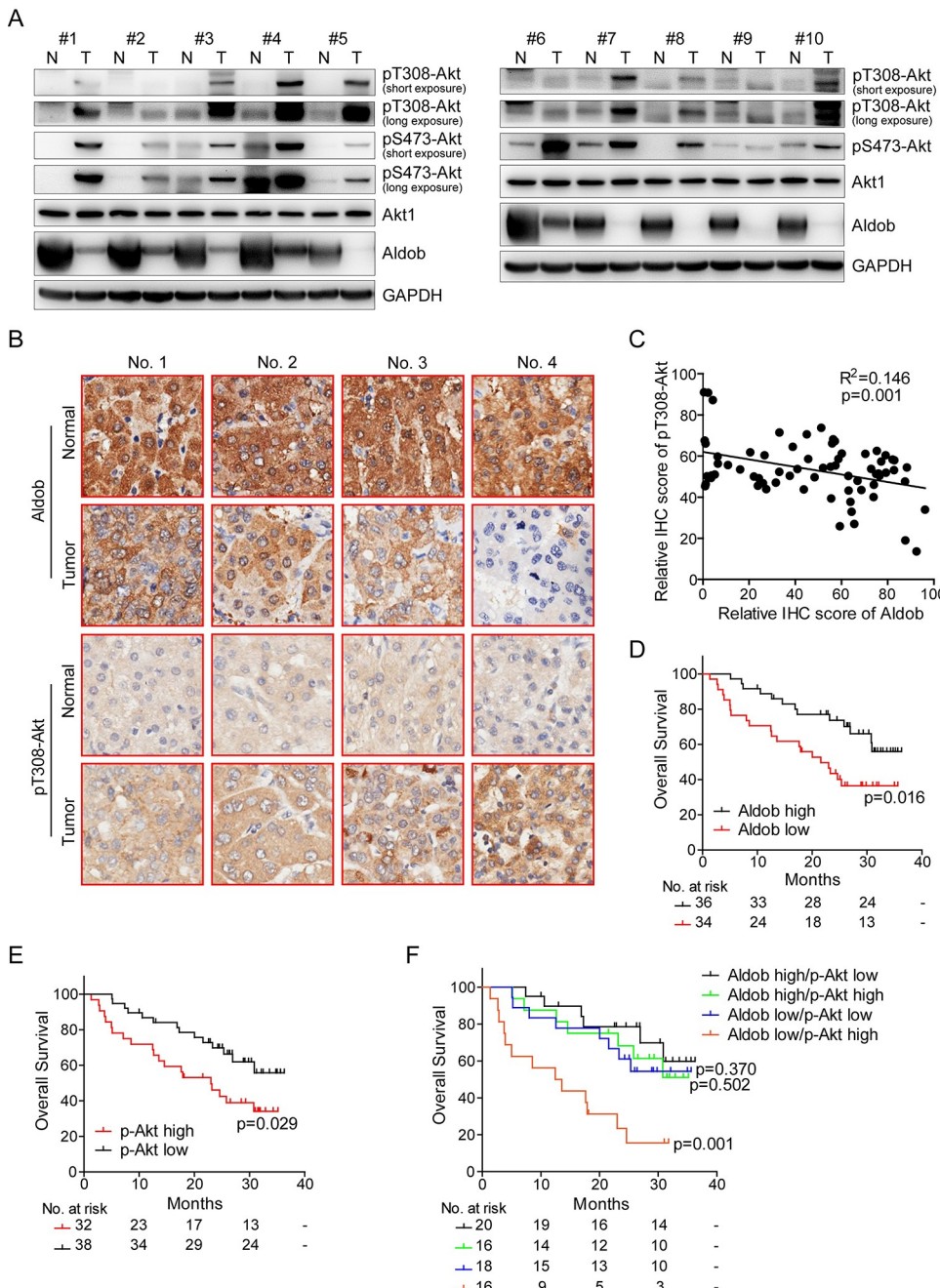

**Fig 1. A negative correlation between Aldob and p-Akt expression in human HCC specimens. (A)** IB analysis of WCLs derived from 10 pairs of human HCC tumors and matched adjacent nonmalignant tissues. **(B)** Representative IHC staining images of Aldob and pT308-Akt from the same HCC patients on tissue microarray were shown (original magnification ×200). **(C)** The relative IHC score of Aldob and pT308-Akt from tissue microarray of 70 HCC patients was plotted and assessed by a linear regression analysis. Protein expression score is classified into negative (0–20), weak (21–100), moderate (101–180), and strong (181–255). The score calculation formula is (weak + moderate + strong)/ (negative + weak + moderate + strong) × 100%. **(D** and **E)** Overall survival of HCC patients ($n = 70$) grouped by high IHC score ($\geq$50) or low IHC score ($<$50) of Aldob **(D)** or pT308-Akt **(E)** expression levels were conducted through the Kaplan–Meier analysis and log-rank test. The number of patients at risk was listed below each curve. **(F)** Overall survival of HCC patients ($n = 70$) grouped by different combination of Aldob and pT308-Akt levels were conducted through the Kaplan–Meier analysis and log-rank test. The number of patients at risk was listed under each curve. The data underlying this figure can be found in S1 Data. Aldob, aldolase B; HCC, hepatocellular carcinoma; IB, immunoblot; IHC, immunohistochemistry; WCL, whole cell lysate.

type (WT) (ALDOB$^{f/f}$) PHs, as evidenced by increased levels of pT308-Akt, pS473-Akt, and Akt downstream targets, including pS9-GSK3β, pT389-S6K, hexokinase 1 (HK1), and HK2, without affecting the expression of Akt, GSK3β, and S6K (Fig 2A). Similarly, we observed that loss of Aldob resulted in elevated expression levels of pT308-Akt, pS473-Akt, pS9-GSK3β, and mechanistic target of rapamycin complex 1 (mTORC1) direct downstream target pT389-S6K in liver tumor tissues of global *ALDOB* KO mice compared to WT mice after diethyl nitrosamine (DEN) treatment (Figs 2B and S1B). However, there was no significant change of the phosphorylation level of mTORC2 downstream target protein kinase Cα (PKCα) (S1B Fig). Notably, the expression of HK1 and HK2 were significantly increased in *ALDOB* KO mice, while HK4 expression was decreased (Figs 2B and S1B). Previous study has demonstrated that HCC cells are metabolically distinct from normal hepatocytes through expressing the high-affinity HK1 and HK2 and inhibiting the low-affinity glucokinase (HK4) to increase glucose uptake and metabolism [21]. These results suggest that loss of Aldob may facilitate glucose metabolism through activation of Akt signaling in HCC. In combination with our recent study that loss of Aldob promotes DEN-induced HCC tumorigenesis in *ALDOB* KO mice [18], we hypothesized that Aldob exhibits a potential tumor-suppressive role through inhibiting oncogenic Akt activity.

To further address the role of Aldob in the regulation of Akt phosphorylation in HCC, we generated stable Aldob-overexpressing liver cancer cell lines Huh7 and LM3 and found that ectopic expression of Aldob resulted in a marked reduction of Akt, GSK3β, and S6K phosphorylation levels (Fig 2C). Conversely, silencing Aldob by small interfering RNA (siRNA) restored Akt phosphorylation and its downstream targets (Fig 2D). Next, we analyzed the kinetics of Akt activation and attenuation in response to insulin (Fig 2E) and epidermal growth factor (EGF) (Fig 2F) stimulation. Down-regulation of Aldob led to increased levels of pT308-Akt and pS473-Akt with coordinate changes in immediate Akt downstream substrate pS9-GSK3β expression in a time-dependent manner (Fig 2E and 2F). Additionally, we performed an in vitro Akt kinase assay and demonstrated that overexpressed Aldob dramatically inhibited Akt kinase activity, as assessed by the decreased phosphorylation level of Akt substrate GSK3α (Fig 2G). Taken together, all these results strongly suggest that Aldob suppresses Akt phosphorylation and its kinase activity in HCC.

## Aldob inhibits Akt kinase activity and suppresses HCC through impeding cell cycle progression and attenuating glycolysis and TCA metabolism

Given that aberrant activation of Akt signaling promotes cancer progression through regulating cell proliferation, metabolism, and survival [3], we next investigated the functional consequence of Aldob-mediated suppression of Akt kinase activity. We observed that enforced expression of Aldob inhibited cell viability and colony formation capability. Consistently, inhibition of Akt activity by MK2206, a highly selective allosteric Akt inhibitor [22], significantly blocked cell proliferation and eliminated the growth inhibitory effects of Aldob expression (Fig 3A and 3B). Furthermore, activation of Akt signaling by ectopic expression of Myc-Akt1 significantly enhanced cell survival and abolished Aldob overexpression-induced cell growth arrest (S2A and S2B Fig). These results indicate that Akt-dependent signaling plays an essential role in Aldob-mediated suppression of HCC cell proliferation. Meanwhile, Aldob overexpression or inhibiting Akt activity by MK2206 significantly increased the proportion of cells in the G1-phase, resulting in a decreased percentage of cells in S-phase, indicating that Aldob induces cell cycle arrest at the G1-to-S transition (Fig 3C). Consistent with this observation, up-regulated Aldob obviously reduced pT308-Akt, pS473-Akt, pS9-GSK3β, and CyclinD1 expression, while elevating the levels of cell cycle inhibitors p21 and p27 (Fig 3D). More importantly,

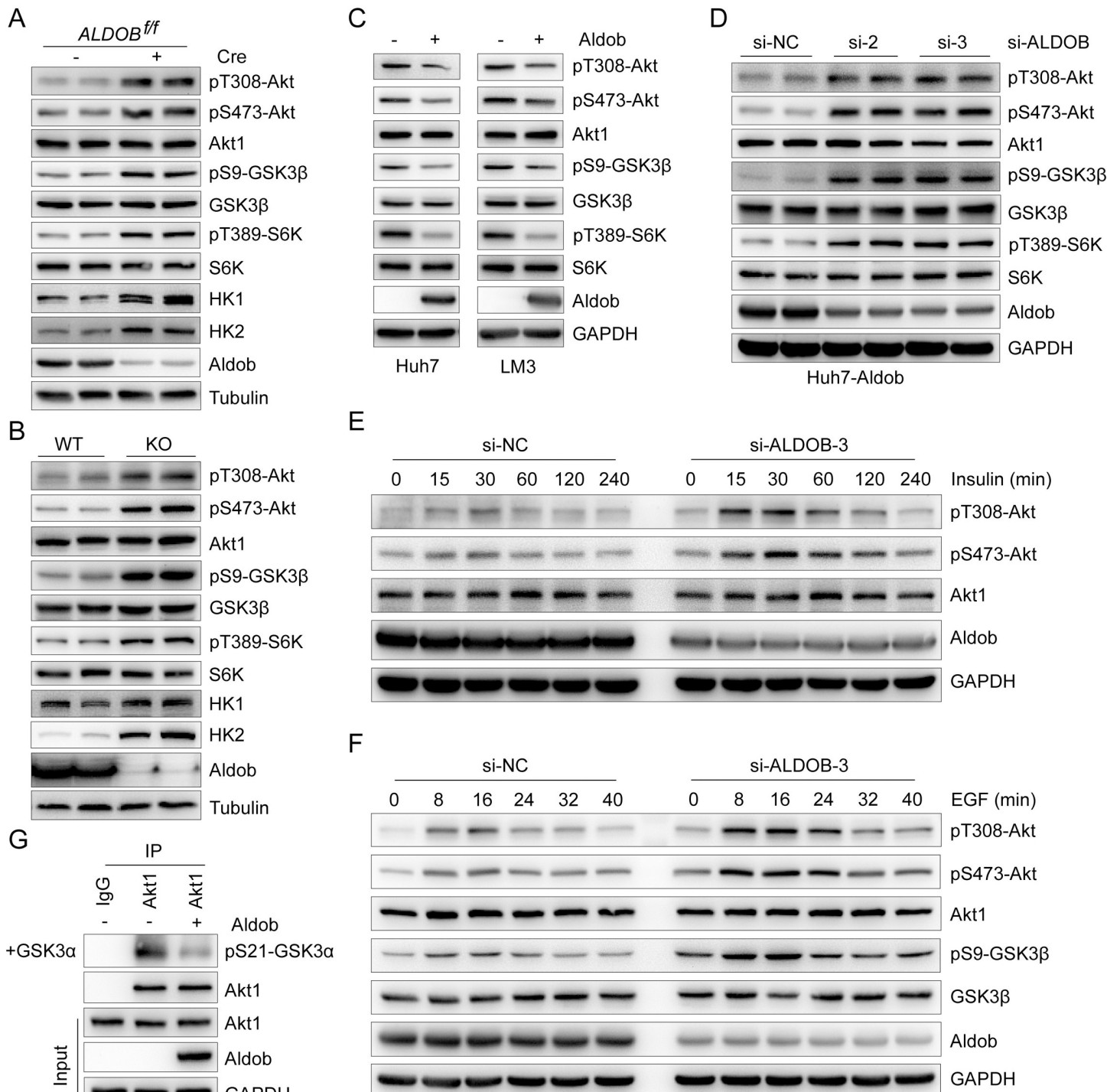

**Fig 2. Aldob suppresses Akt phosphorylation and kinase activity. (A)** IB analysis of WCL derived from conditional *ALDOB* KO and WT mouse primary hepatocytes. **(B)** IB analysis of WCL derived from liver tumor tissues of *ALDOB* KO and WT mice after injection with DEN at postnatal day 14 to induce hepatocellular carcinoma for 10 months. **(C)** IB analysis of WCL derived from Huh7 and LM3 cells stably expressing Aldob via lentiviral infection (with Vector as a negative control). **(D)** IB analysis of WCL derived from Huh7-Aldob cells transfected with indicated siRNAs (si-NC as a negative control). **(E and F)** IB analysis of WCL derived from Huh7-Aldob cells transfected with indicated siRNAs. Resulting cells were serum-starved for 24 hours and then stimulated with 0.1 μM insulin **(E)** or 100 ng/ml EGF **(F)** at the indicated time before harvesting. **(G)** In vitro Akt1 kinase assay was conducted with recombinant GSK3α as a substrate and immunoprecipitated Akt1 from Huh7-Vector or Huh7-Aldob cells as the source of kinase. IgG was used as a negative control. The data underlying this figure can be found in S1 Data. Aldob, aldolase B; DEN, diethyl nitrosamine; EGF, epidermal growth factor; GSK3α, glycogen synthase kinase 3α; Huh7-Aldob, Aldob-overexpressing Huh7; IB, immunoblot; IgG, immunoglobulin G; KO, knockout; siRNA, small interfering RNA; WCL, whole cell lysate; WT, wild-type.

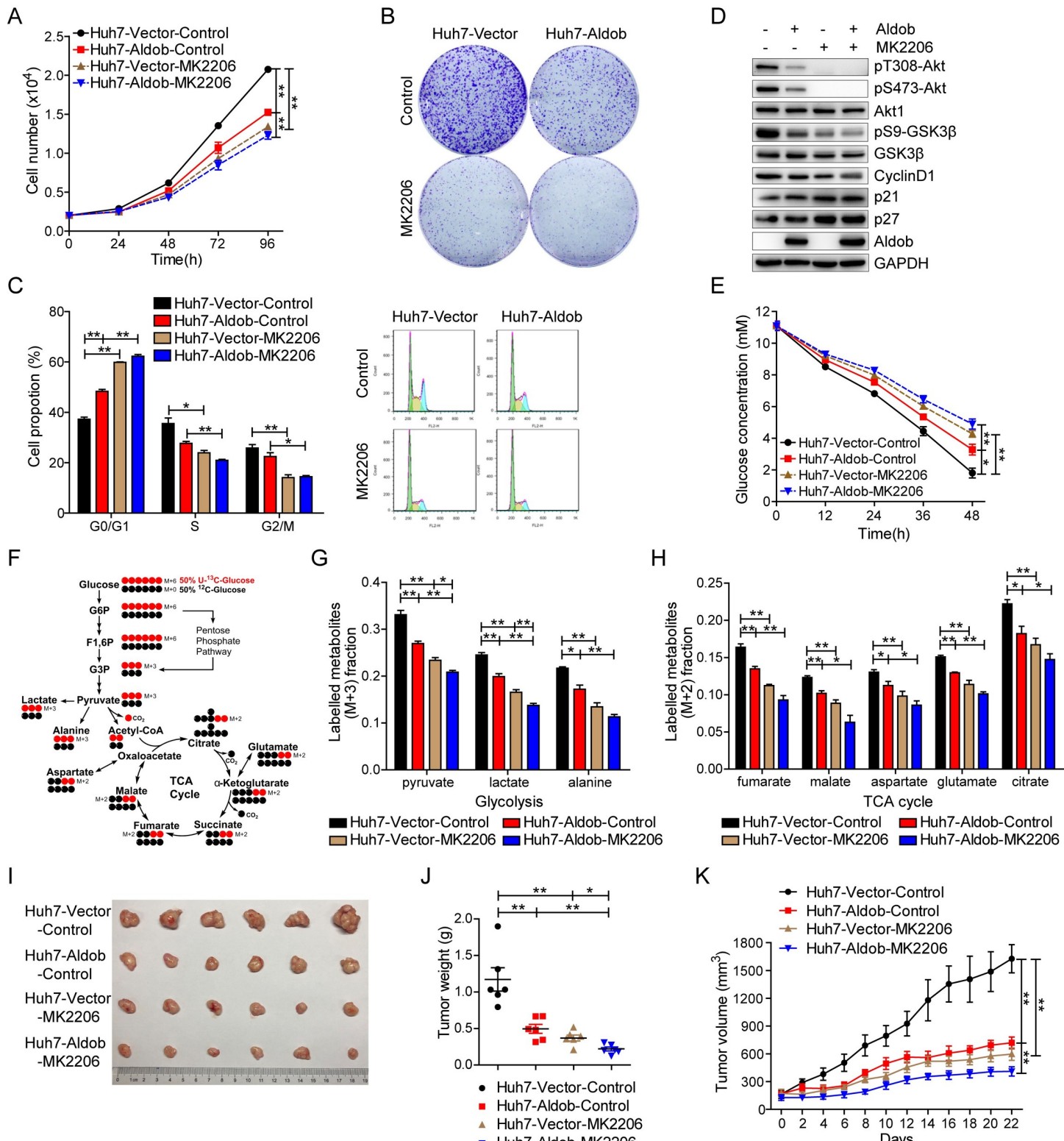

**Fig 3. Aldob-mediated inhibition of Akt activity is required for Aldob-induced tumor-suppressive effects. (A)** Huh7-Vector and Huh7-Aldob cells were treated with DMSO or MK2206 (2 μM) at the indicated time and cell viability was measured by CCK-8 assays. **(B)** Huh7-Vector and Huh7-Aldob cells (5,000 cells/well in 6-well plates) were cultured with or without MK2206 (2 μM) for 10 days before being fixed and stained. Representative graphs of cell colonies were shown. **(C and D)** Cells were treated with DMSO or MK2206 (2 μM) for 48 hours, and then monitored for cell cycle distribution by FACS **(C)** or subjected to IB analysis with the indicated antibodies **(D)**. **(E)**

Glucose levels in the culture medium of Huh7-Vector and Huh7-Aldob cells after treatment with DMSO or MK2206 (5 μM) at different time points. **(F)** Schematic presentation of $^{13}$C distribution in glycolysis and the first turn of the TCA cycle with 50% U-$^{13}$C-glucose (labeled at all 6 carbons, red circles) and 50% $^{12}$C-glucose (unlabeled, black circles). **(G and H)** Fraction of the labeled metabolites of M+3 from $^{13}$C-glucose in glycolysis **(G)** and fraction of the labeled metabolites of M+2 from $^{13}$C-glucose in TCA cycle **(H)** by DMSO or MK2206 (5 μM) treatment for 12 hours in Huh7-Vector and Huh7-Aldob cells. **(I–K)** Huh7-Vector and Huh7-Aldob cells were injected subcutaneously into the flanks of nude mice, followed by treatment with Control solvent or MK2206 ($n$ = 6 mice). Representative tumor images **(I)**, xenograft tumor weight **(J),** and tumor size **(K)** in each group were recorded. Data are presented as mean ± SEM. $^*$ $p < 0.05$; $^{**}$ $p < 0.01$ (Student $t$ test). The data underlying this figure can be found in S1 Data. Aldob, aldolase B; CCK-8, Cell Counting Kit-8; FACS, fluorescence-activated cell sorting; F1,6P, fructose-1,6-bisphosphate; G3P, glyceraldehyde-3-phosphate; G6P, glucose-6-phosphate; Huh7-Aldob, Aldob-overexpressing Huh7; IB, immunoblot; TCA cycle, tricarboxylic acid cycle.

either Akt inhibition using MK2206 or Akt activation by ectopic expression of Myc-Akt1, but not the Myc-Akt1 phospho-deficient mutant (Myc-Akt1-AA), dramatically diminished Aldob-mediated abovementioned tumor-suppressive effects (Figs 3C and 3D and S2C–S2F).

Metabolic reprogramming has been recognized as the core hallmark for various cancers [23]. Enhanced aerobic glycolysis, known as the Warburg effect, is frequently observed along with abnormally high rates of glucose uptake in rapidly proliferating cancer cells [24]. As the central route of oxidative phosphorylation, the tricarboxylic acid (TCA) cycle is also up-regulated to meet increased cellular energy, biosynthesis, and redox needs [25,26]. The downstream effectors of Akt signaling play a central role in cancer cell metabolic reprogramming through regulating glucose uptake and metabolism. To further determine the potential roles of Aldob-mediated inhibition of Akt activity on glucose metabolism, we observed that glucose levels were time-dependently decreased in the cell culture medium (Fig 3E). Interestingly, Huh7-Vector cells depleted glucose from the culture medium more efficiently than Huh7-Aldob cells. Furthermore, inhibition of Akt activity by MK2206 significantly suppressed glucose consumption and mitigated Aldob overexpression-induced inhibitory effects on glucose consumption (Fig 3E). Next, we used stable isotope labeled [U-$^{13}$C$_6$] glucose as a tracer to track intracellular metabolic flux (Fig 3F). The M+3 ($^{13}$C labeled at all 3 positions) fractions of enriched labeled metabolites in glycolysis including pyruvate, lactate, and alanine were significantly decreased in Huh7-Aldob cells compared with control (Fig 3G). Meanwhile, Aldob overexpression also observably reduced the M+2 ($^{13}$C labeled at 2 positions) fractions of labeled metabolites in TCA cycle, such as fumarate, malate, aspartate, glutamate, and citrate (Fig 3H). Besides, blocking Akt activity by MK2206 efficiently weakened Aldob-induced inhibition of metabolic flux in glycolysis and TCA cycle (Fig 3G and 3H). These results suggest that Aldob suppresses Akt activity, resulting in the reduction in glucose consumption and cellular metabolism in glycolysis and TCA.

On the other hand, endoribonuclease-prepared siRNA (esiRNA)-mediated knockdown of endogenous Aldob expression in Huh7 cells significantly promoted tumor cell proliferation, glucose consumption, lactate production, and activation of Akt signaling (S3A–S3D Fig). Conversely, Akt inhibition using MK2206 reversed Aldob knockdown-induced oncogenic properties (S3A–S3D Fig). Furthermore, knockdown of Aldob via siRNA in Huh7-Aldob cells markedly enhanced Akt activity, promoted cell proliferation and cell cycle progression, increased glucose consumption and the levels of metabolites in glycolysis and TCA cycle, and rescued overexpressed Aldob-mediated inhibition of cell growth and metabolism (S4A–S4J Fig). Moreover, MK2206 exposure also reversed the growth-promoting effects of Aldob knockdown in Huh7-Aldob cells (S4A–S4J Fig). We found similar patterns of cell proliferation and cell cycle progression effects in LM3 cells (S5A–S5D Fig). These results suggest that Aldob impedes cell cycle progression and represses glucose metabolism through inhibiting Akt signaling, which contributes to Aldob-induced cell growth suppression.

We then established a subcutaneous xenograft mouse model to investigate the effects of Aldob expression on tumorigenesis in vivo. As illustrated in Fig 3I–3K, tumors derived from

Huh7-Aldob cells exhibited slower growth and smaller sizes than those from Huh7-Vector cells. Inhibition of Akt activity by MK2206 alone significantly inhibited tumor growth. Additionally, a combination of Aldob overexpression and inhibition of Akt activity showed an additive effect on tumor growth inhibition. Interestingly, Huh7-Vector cells appeared to be more responsive to MK2206 than Huh7-Aldob cells in which Akt signaling was already suppressed at baseline. Moreover, Ki67 expression was consistent with the differences noted in tumor weights and size (S6A Fig). Likewise, western blot analysis revealed that Aldob overexpression and MK2206 treatment alone significantly inhibited Akt activation and its downstream signaling compared to control group, respectively, and the combination of the two had the lowest levels of Akt signaling-related proteins (S6B Fig). No significant changes in body weight were observed throughout the study (S6C Fig). Together, these data demonstrate that Aldob exhibits tumor suppressive effects through inhibition of Akt signaling in vitro and in vivo; inhibition of Akt activity by the allosteric inhibitor suppresses tumorigenesis caused by the loss of Aldob.

## Aldob directly interacts with Akt to suppress Akt activity and HCC cell growth, whereas disruption of Aldob/Akt interaction in Aldob mutant cells restores Akt activity and cell proliferation

Next, we tested whether Aldob modulated Akt activity through a protein–protein interaction by performing immunoprecipitation (IP) experiments. As shown in Fig 4A, endogenous Aldob pulled down Akt1 in WT mice liver tissues. Reciprocally, Aldob was detected in the endogenous Akt1 immunocomplex. However, *ALDOB* KO significantly weakened immunoprecipitated interacting proteins, suggesting that Aldob and Akt1 bind with each other. Consistently, the interactions between overexpressed Aldob and endogenous or exogenous Akt1 were observed by co-IP assays in HCC cell lines (Figs 4B and 4C and S5E). Knockdown of Aldob or Akt1 expression by siRNA greatly decreased Aldob interaction with Akt1 (Fig 4B). Moreover, we detected a direct interaction between the recombinant glutathione S-transferase (GST)-Akt1 and His-Aldob performed by GST pull-down analysis (Fig 4D). In addition, Aldob and Akt2 combined with each other (S7A Fig). Collectively, these results demonstrate that Aldob physically interacts with Akt.

Hepatic Aldob is indispensable for glucose and fructose metabolism, and its deficiency as result of *ALDOB* gene mutation leads to HFI in humans [27]. The residues at Arg43, Arg46, Lys108, Lys147, Arg149, Lys 230, and Arg304 line within the active site pocket of Aldob, and play an important role in substrate binding and catalysis [28,29]. Mutations involved in aforementioned residues have been identified in the *ALDOB* gene of HFI patients [30,31]. To determine whether the enzymatic activity of Aldob is required for this novel interaction, we individually mutated the 7 positively charged amino acid residues to alanine and found that among the 4 point mutants (R43A, K108A, K230A, and R304A) with considerably reduced Aldob-Akt interaction, they exhibited varied Aldob enzymatic activities: R43A and K230A (completely dead), K108A (32.1%), and R304A (49.1%) of WT Aldob (Fig 4E). Among them, R304A mutant has the most significantly attenuated interactions with Akt. Moreover, the other 3 mutants (R46A, K147A, and R149A) with moderately attenuated Aldob-Akt interaction also had variable enzyme activities: R46A maintained 60.2% activity and the other two did not have significant enzyme activity (Fig 4E). Together, these data clearly indicate that the interaction of Aldob with Akt is independent of Aldob enzymatic activity.

To further identify the Akt interacting region on Aldob, a full-length and 2 truncated mutants of Aldob including a.a. 1–155 (amino acid 1–155 fragment containing clone) and a.a. 1–240 were constructed to examine their Akt-binding capability. Intriguingly, both Aldob truncated mutants failed to interact with Akt compared to the full-length Aldob, suggesting

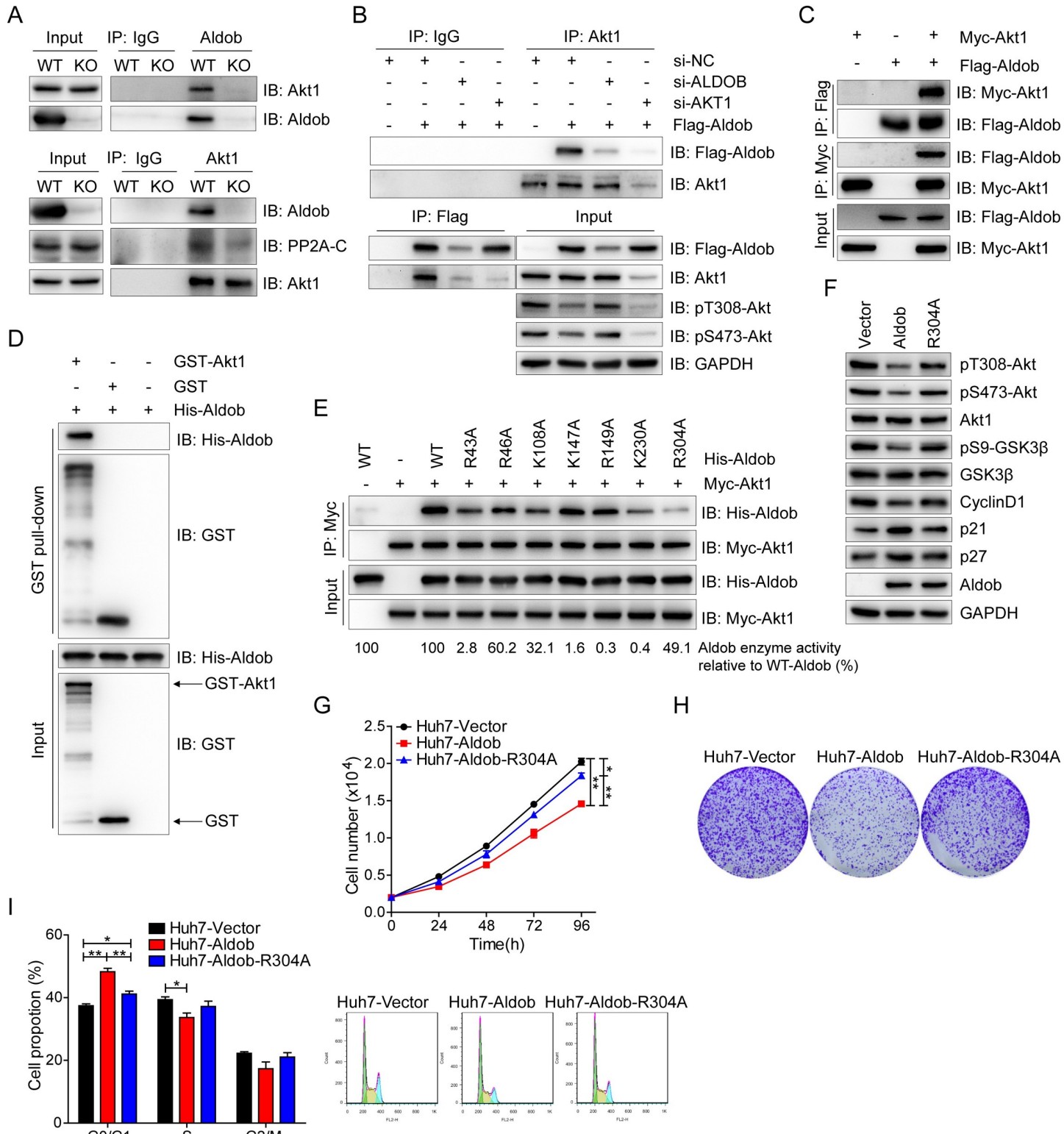

**Fig 4. The direct interaction of Aldob with Akt is essential for Aldob to suppress Akt activity and HCC cell growth.** (**A**) IB analysis of endogenous anti-Aldob and anti-Akt1 co-IP and WCL derived from liver tissues of *ALDOB* KO and WT mice. IgG was used as a negative control. (**B**) IB analysis of anti-Akt1 and anti-Flag IP and WCL derived from Huh7 cells transfected with the indicated constructs and siRNAs. IgG was used as a negative control. (**C**) IB analysis of co-IP and WCL derived from Huh7 cells transfected with the indicated constructs. (**D**) GST pull-down analysis to demonstrate the direct interaction between recombinant proteins GST-Akt1 and His-Aldob. Recombinant GST protein was used as a negative control. (**E**) IP analysis was performed with WCL derived from Myc-Akt1-transfected Huh7 cells and

recombinant protein WT-His-Aldob or their various His-Aldob mutants harboring a lysine or arginine to alanine mutation in the identified active site residues of Aldob. The Aldob enzyme activity in Aldob mutants relative to WT-His-Aldob were calculated (bottom). **(F)** IB analysis of WCL derived from Huh7 cells transfected with WT or R304A mutant form of Flag-Aldob constructs. **(G–I)** Huh7 cells transfected with the indicated constructs were used to determine their effects on cell proliferation **(G)**, colony formation **(H),** and cell cycle distribution **(I)**. Data are presented as mean ± SEM. * $p < 0.05$; ** $p < 0.01$ (Student $t$ test). The data underlying this figure can be found in S1 Data. Aldob, aldolase B; co-IP, co-immunoprecipitation; GST, glutathione S-transferase; HCC, hepatocellular carcinoma; IB, immunoblot; IgG, immunoglobulin G; IP, immunoprecipitation; KO, knockout; siRNA, small interfering RNA; WCL, whole cell lysate; WT, wild-type.

that the carboxyl-terminal region of Aldob (a.a. 241–364) is most likely the region for Akt interaction (S7B Fig), consistent with the observation that R304A has the weakest Aldob-Akt interaction. We next tested whether Aldob-Akt interaction is essential for Aldob regulation of Akt activity and HCC cell growth. Overexpression of WT Aldob significantly down-regulated Akt signaling, whereas disruption of Akt/Aldob interaction in R304A mutant abrogated Aldob-induced inhibition of Akt signaling (Fig 4F). Consistently, R304A mutant efficiently rescued cell viability, colony formation, and cell cycle progression in Aldob-overexpressing cells (Fig 4G–4I). Together, these data indicate that Aldob directly interacts with Akt to suppress Akt activity and that disruption of this interaction releases Akt kinase activity to promote cancer cell growth.

## Aldob preferentially associates with Akt in a phosphorylation-dependent manner

Because the state of Akt phosphorylation is critical for the integration of various extracellular cues to downstream signaling events for multiple biological processes, we next studied whether the phosphorylation state of Akt played any role in the regulation of the protein–protein interaction between Akt and Aldob. To this end, we examined the effect of Akt1 phospho-deficient mutants (Akt1-T308A and Akt1-S473A) on Aldob-binding capability. Compared with Akt1-WT, both Akt1 phospho-deficient mutants displayed reduced binding with Aldob, suggesting that Akt1 kinase activity is critical for Akt1 and Aldob interaction (Fig 5A). Next, we treated Aldob- and Akt1-transfected Huh7 cells with insulin or EGF to further examine the correlation between Akt1 phosphorylation status and Aldob interaction with Akt1. Interestingly, the interaction of Aldob and Akt1 tracked with the kinetics of Akt1 activation and attenuation in response to either insulin or EGF, with the binding between Aldob and Akt1 almost peaking when the levels of phosphorylated Akt (p-Akt) were the highest (Fig 5B and 5C). Furthermore, blocking Akt phosphorylation through MK2206 treatment prominently decreased Aldob interaction with Akt1 (Fig 5D). Together, these results indicate that Aldob preferentially binds to the phosphorylated form of Akt.

## Aldob promotes Akt interaction with PP2A and facilitates PP2A-mediated Akt dephosphorylation

Given that Akt interacted with Aldob in an Akt phosphorylation-dependent manner, we hypothesized that phosphatases of Akt may be involved in Aldob-induced repression of Akt activity. Interestingly, as shown in Fig 2, Akt phosphorylation at both T308 and S473 was significantly reduced in response to Aldob expression. Previous study suggested that PP2A directly dephosphorylates Akt at both T308 and S473 [32,33]. Moreover, phosphorylation of Akt at T308 has been identified to play a more crucial role in activating Akt [34,35]. Thus, we examined the possible involvement of the serine/threonine phosphatase PP2A in Aldob-mediated regulation of Akt activity. Strikingly, we observed an abundance of the catalytic subunit of PP2A (PP2A-C), but little PHLPP, in the exogenous Aldob immunocomplex (Fig 6A).

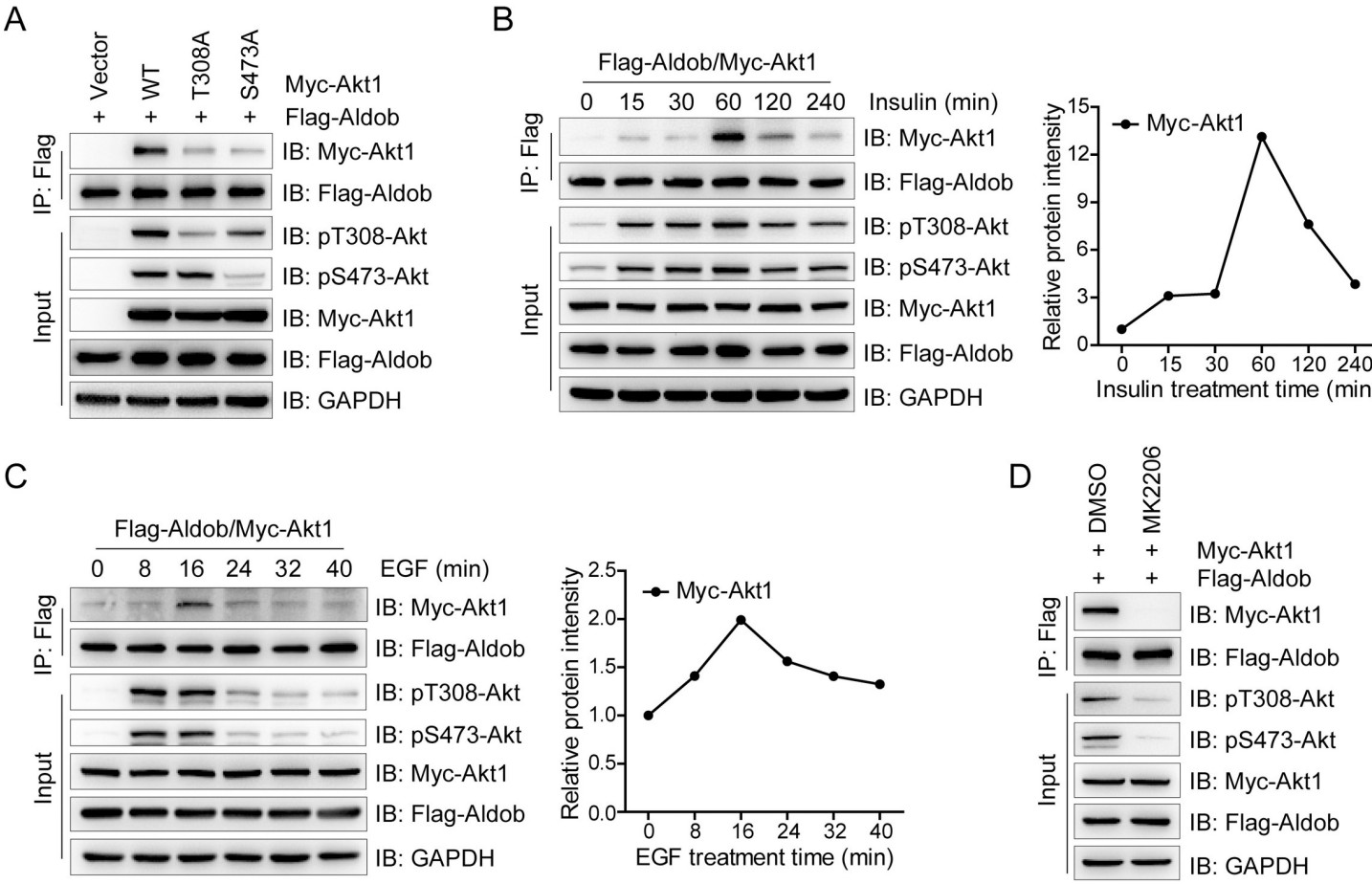

**Fig 5. Aldob preferentially interacts with the phosphorylated species of Akt1. (A)** IB analysis of anti-Flag IP and WCL derived from Huh7 cells transfected with the indicated constructs. T308A and S473A indicated that threonine at position 308 and serine at position 473 of Akt1 were replaced by alanine, respectively. **(B** and **C)** IB analysis of anti-Flag IP and WCL derived from Huh7 cells transfected with Flag-Aldob and Myc-Akt1. Resulting cells were serum-starved for 24 hours and then stimulated with 0.1 μM insulin **(B)** or 100 ng/ml EGF **(C)**. At the indicated time points, WCL were harvested for Flag-IP and further for IB analysis (left). The relative intensity of Myc-Akt1 immunoprecipitated by Flag-Aldob was quantified (right). **(D)** IB analysis of anti-Flag IP and WCL derived from Flag-Aldob and Myc-Akt1 transfected Huh7 cells that were treated with DMSO or MK2206 (1 μM) for 24 hours before harvesting. The data underlying this figure can be found in S1 Data. Aldob, aldolase B; EGF, epidermal growth factor; IB, immunoblot; IP, immunoprecipitation; WCL, whole cell lysate.

Notably, neither interaction of Aldob with the Akt kinase PDK1, the structural subunit (PP2A-A), nor the regulatory subunit (PP2A-B55α) of PP2A was observed (Figs 6B and S7C).

Since PP2A has been shown to directly bind to Akt and dephosphorylate Akt [36], we speculated that Aldob might nucleate the Akt interaction with PP2A to trigger Akt dephosphorylation resulting in inhibition of downstream signaling. To support this hypothesis, we performed GST pull down assay and found that WT Aldob, but not the Akt1-binding deficient Aldob-R304A mutant, significantly enhanced Akt1 interaction with PP2A-C (Fig 6C). Moreover, IP analysis showed that Aldob overexpression markedly facilitated PP2A-C interaction with Akt1 and Aldob without a significant impact on the interactions between PP2A-C and PP2A-A or B55α subunits (S7D Fig). Similarly, endogenous Akt1 pulled down both Aldob and PP2A-C in the liver tissue of WT mice, whereas loss of Aldob drastically diminished Akt1-immunoprecipitated Aldob and PP2A-C in *ALDOB* KO mice (Fig 4A), further suggesting that Aldob acts as a scaffold protein for the Akt/PP2A complex. Next, we performed in vitro PP2A phosphatase activity assay using phosphorylated Akt as a substrate and observed a time-

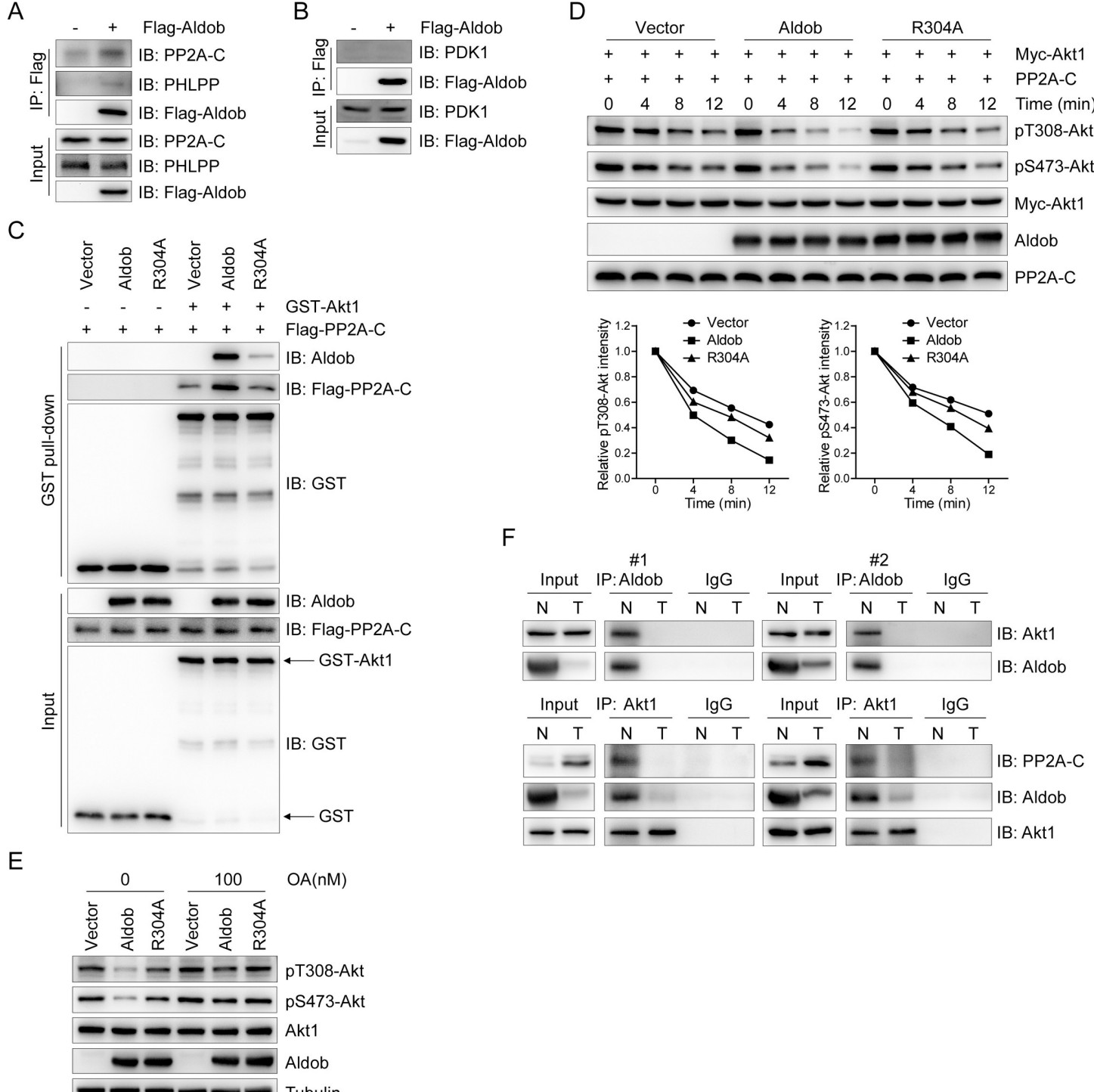

**Fig 6. Aldob regulates Akt activity through protein phosphatase PP2A. (A** and **B)** IB analysis of anti-Flag IP and WCL derived from Huh7 cells transfected with the indicated constructs. **(C)** GST pull-down assays were performed with WCL derived from Flag-PP2A-C-transfected Huh7 cells and recombinant proteins GST-Akt1 and WT-Aldob or Aldob-R304A mutant. GST was used as the negative control. **(D)** In vitro PP2A dephosphorylation assays were performed with phosphorylated Myc-Akt1 and active PP2A in the presence or absence of recombinant WT-Aldob or Aldob-R304A mutant (top). The relative signal intensities of pT308-Akt and pS473-Akt were quantified (bottom). **(E)** IB analysis of WCL derived from Huh7 cells transfected with the indicated constructs. Resulting cells were treated for 4 hours with OA at 100 nM. **(F)** IB analysis of endogenous anti-Aldob and anti-Akt1 IP and WCL derived from 2 pairs of human HCC samples. IgG was used as the negative control. The data underlying this figure can be found in S1 Data. Aldob, aldolase B; IB, immunoblot; IP, immunoprecipitation; GST, glutathione S-transferase; IgG, immunoglobulin G; OA, okadaic acid; PP2A, protein phosphatase 2A; WCL, whole cell lysate; WT, wild-type.

dependent decrease of p-Akt at both T308 and S473 residues upon incubation with active PP2A (Fig 6D). More importantly, WT Aldob remarkably augmented the PP2A-induced reduction of p-Akt expression, while no significant effect was found on the Akt1-binding deficient Aldob-R304A mutant (Fig 6D). Furthermore, we examined the effect of okadaic acid (OA) on these processes. OA is an inhibitor of serine/threonine protein phosphatase PP1 and PP2A, exhibiting greater selectivity toward the latter up to concentration of 100 nM [37]. As shown in Fig 6E, OA treatment effectively restored Akt phosphorylation levels in Aldob-over-expressing cells, which almost comparable to untreated control cells, suggesting that Aldob-induced inhibition of Akt activity is dependent on PP2A. To further verify these results in human HCC samples, we randomly selected 2 pairs of HCC tissue samples for IP assays. The endogenous interaction between Aldob and Akt1 was identified in peripheral normal liver tissues of HCC patients (Fig 6F). However, down-regulated Aldob in tumor tissues significantly decreased Akt1-immunoprecipitated Aldob and PP2A-C when compared to corresponding adjacent normal tissues (Fig 6F), further suggesting that Aldob/Akt/PP2A signaling is involved in human HCC progression. Taken together, Aldob interacts with p-Akt to promote the recruitment of the phosphatase PP2A and accelerate PP2A-mediated dephosphorylation, which is primarily responsible for Aldob-induced down-regulation of Akt phosphorylation.

## PP2A activation using SMAP down-regulates Akt activity and suppresses HCC progression

PP2A is generally considered as a tumor suppressor that dephosphorylates multiple critical oncogenic proteins, such as Akt, extracellular signal-regulated kinase (ERK), and MYC [38]. Functional inactivation of PP2A has been linked to tumor development in many cancers [39]. Thus, pharmacologic restoration of PP2A phosphatase activity has emerged as an attractive strategy for cancer therapy [40]. Recent studies have reported that a series of specific SMAPs reengineered from tricyclic neuroleptics effectively activate PP2A, resulting in the dephosphorylation of key targets Akt and ERK, and blocking tumor growth of lung, prostate cancer, and pancreatic neuroendocrine tumors both in vitro and in vivo [41–44]. The validation of their target specificity, mechanism of action, and pharmaceutic properties have been extensively documented [41–44]. Here, we treated Huh7 and LM3 cells with increasing concentrations of SMAP. SMAP-induced PP2A activation was demonstrated by parallel reduction of Akt, ERK, and GSK3β phosphorylation levels, which was accompanied by decreased CyclinD1 expression (S8A Fig). Moreover, SMAP treatment reduced cell viability in both cell lines in a dose-dependent manner with $IC_{50}$ of approximately 9 μM (Huh7) and 12 μM (LM3), similar to those reported in other tumor cell types (S8B Fig). SMAP treatment induced visible HCC cell death. These results indicate that PP2A activation with SMAP results in growth inhibition and cell death in HCC cell lines.

To evaluate the utility of PP2A activation toward Aldob-mediated suppression of Akt activity, Huh7-Vector and Huh7-Aldob cells were treated with 10 μM SMAP for further functional studies. We found that PP2A activation with SMAP effectively reversed the promoting effects of esi-RNA-mediated Aldob knockdown on cell proliferation, glucose consumption, lactate production, and activation of Akt signaling (Fig 7A–7D). Furthermore, Aldob overexpression or SMAP treatment alone inhibited cell proliferation, and the combination of the two showed an additive effect on repressing cell proliferation, as measured by cell viability and clonogenic assays (Fig 7E and 7F). Additionally, cell cycle analysis revealed that SMAP treatment resulted in a significant increase in the G1 phase with a coordinate loss of the S population, and exacerbated Aldob-induced G1-to-S phase cell cycle arrest (Fig 7G). Western blot analysis showed Aldob overexpression had no effect on ERK phosphorylation in the presence of either DMSO

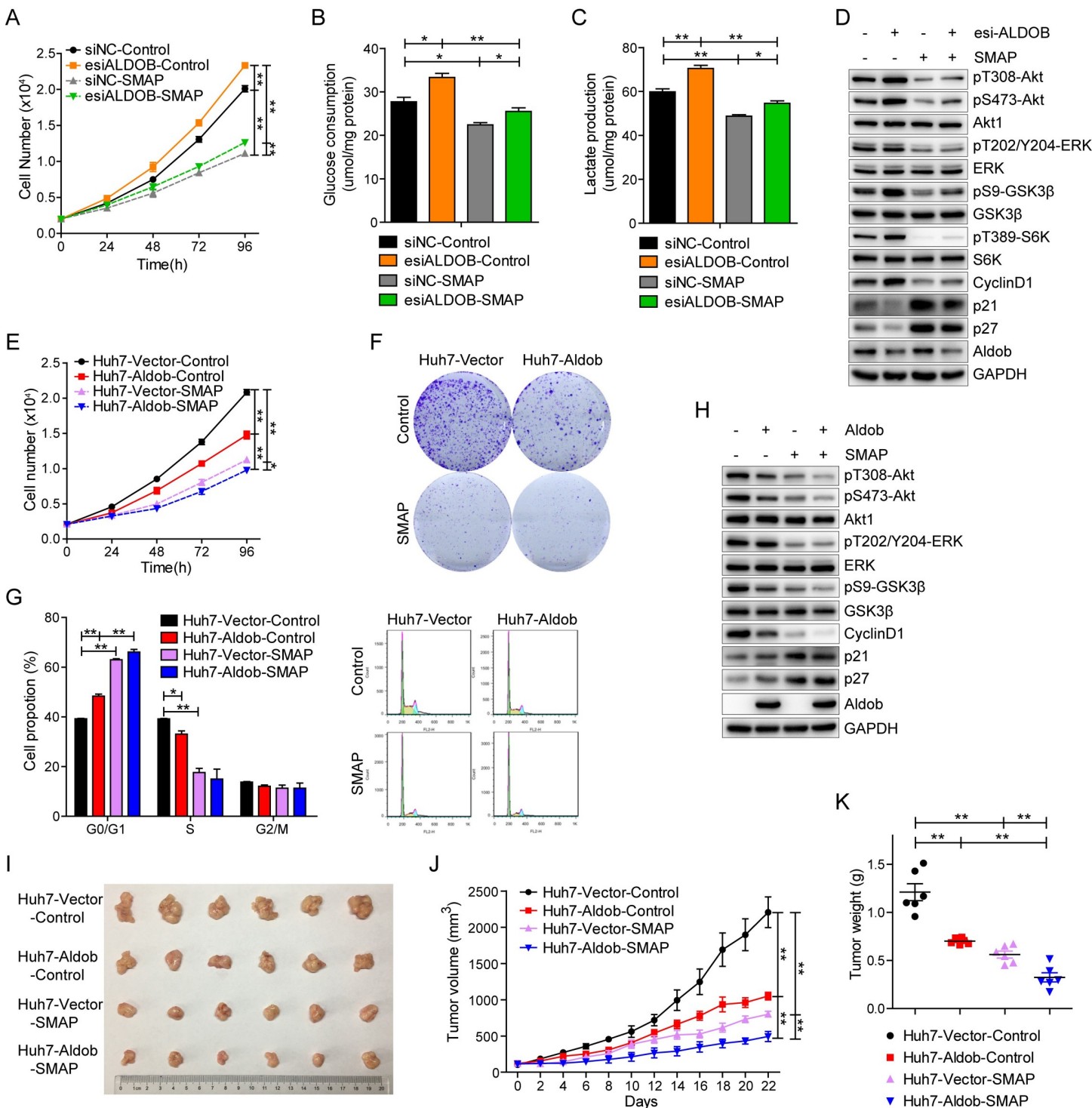

**Fig 7. Activation of PP2A with SMAP potentiates Aldob-mediated tumor-suppressive role. (A–D)** Huh7 cells were transfected with esiRNA for knockdown of endogenous Aldob, and cell proliferation **(A)**, glucose consumption **(B)**, lactate production **(C),** and the protein expression levels in Akt signaling pathway **(D)** were determined in the presence of control DMSO or SMAP (10 μM). **(E)** Cell proliferation of Huh7-Vector and Huh7-Aldob cells in the presence of DMSO or SMAP (10 μM). **(F)** Representative graphs from colony formation assay of Huh7-Vector and Huh7-Aldob cells after exposure to DMSO or SMAP (10 μM) for 14 days. **(G)** Huh7-Vector and Huh7-Aldob cells were treated with 10 μM SMAP for 24 hours, and cell cycle distribution was monitored by FACS. **(H)** IB analysis of WCL derived from Huh7-Vector and Huh7-Aldob cells treated for 20 hours with control DMSO or SMAP (10 μM). **(I–K)** Representative tumor images **(I)**, xenograft tumor volume **(J),** and tumor weight **(K)** of Huh7-Vector and Huh7-Aldob xenograft tumors from control solvent and SMAP-treated nude mice (*n* = 6 mice). Data are presented as mean ± SEM. * *p* < 0.05; ** *p* < 0.01 (Student *t* test). The data underlying this figure can be found in S1 Data. Aldob, aldolase B; esiRNA, endoribonuclease-prepared siRNA; FACS, fluorescence-activated cell sorting; IB, immunoblot; PP2A, protein phosphatase 2A; SMAP, small-molecule activator of PP2A; WCL, whole cell lysate.

or SMAP, whereas Aldob-induced down-regulation of Akt/GSK3β/CyclinD1 signaling was potentiated by SMAP-mediated PP2A activation (Fig 7H). Notably, relative to Huh7-Aldob cells that contain diminished Akt activity, Huh7-Vector cells were more sensitive to SMAP treatment. Together, these results indicate that PP2A plays an essential role in Aldob-induced growth inhibition and that small molecule mediated PP2A activation is beneficial to Aldob-mediated suppression of Akt signaling and HCC cell growth.

We further investigated the efficacy of SMAP on HCC tumor progression in vivo. SMAP treatment alone resulted in dramatic inhibition of tumor growth, similar to Aldob overexpression-induced growth-inhibitory effects, whereas overexpression of Aldob and SMAP treatment achieved additive tumor suppressive effects (Fig 7I–7K). It is noteworthy that there was no significant weight loss or behavioral abnormalities during the 3-week treatment, suggesting that SMAP is well tolerated in this mouse model (S8C Fig). In addition, western blot analysis in SMAP-treated tumors detected the simultaneous decrease in Akt/GSK3β/CyclinD1 signaling and ERK phosphorylation levels, suggesting that SMAP induces PP2A-dependent inhibition of these signaling pathways resulting in suppression of tumor growth (S8D Fig). Immunohistochemistry (IHC) analysis validated lower protein levels of Ki67 in SMAP-treated tumors compared with control vehicle-treated tumors (S8E Fig). More importantly, up-regulated Aldob-induced attenuation of Akt signaling and tumor growth was greatly enhanced in response to SMAP treatment (Figs 7I–7K and S8D and S8E). Collectively, these results support the potential therapeutic application of PP2A phosphatase reactivation for the treatment of HCC with Aldob deficiency.

## Discussion

Emerging studies have documented hepatic Aldob deficiency in HCC, but the underlying mechanisms in the context of metabolic reprogramming remain poorly defined. Our recent study uncovered a mechanism by which loss of Aldob led to a novel mode of metabolic reprogramming through up-regulation of glycolysis, PPP, and TCA to promote HCC [18]. This tumor promoting effect due to the loss of Aldob is achieved by releasing the inhibition on G6PD and PPP metabolism as a result of destabilizing Aldob/G6PD/p53 protein complex [18]. However, the upstream signaling events leading to the enhanced glycolysis due to Aldob deficiency are still unknown. In this study, we observe another nonenzymatic tumor-suppressive role of Aldob through a direct interaction with Akt in HCC, independent of Aldob enzymatic activity (Fig 8). Aldob inhibits Akt activity and downstream signaling events, which leads to cell cycle arrest and the impairment of glucose consumption and metabolic flux to glycolysis and TCA cycle, thereby suppressing HCC cell proliferation and tumorigenesis. Furthermore, Aldob interacts with p-Akt to facilitate the recruitment of protein phosphatase PP2A and PP2A-mediated dephosphorylation, resulting in attenuated Akt activity. On the other hand, loss of Aldob or disruption of the Aldob/Akt interaction in Aldob mutants restores Akt activation, leading to cell cycle progression and up-regulation of glucose metabolism through hexokinases expression. Consistently, a reverse correlation of Aldob and p-Akt level is observed in human HCC tumor tissues and a combination of low Aldob expression with high p-Akt expression predicts the worst prognosis in HCC. Importantly, chemical inhibition of Akt kinase with MK2206 or reactivation of PP2A phosphatase activity with specific SMAP represses the tumorigenic effects resulting from the loss of Aldob in vitro and in vivo, further supporting the therapeutic potential of Akt inhibitors or PP2A activators for HCC treatment.

Dysregulated PI3K/Akt pathway contributes to multiple malignancies, and it is of great clinical significance to characterize the molecular mechanisms leading to the hyperactivation of Akt signaling in carcinogenesis. In addition to the hepatitis virus infection and genomic

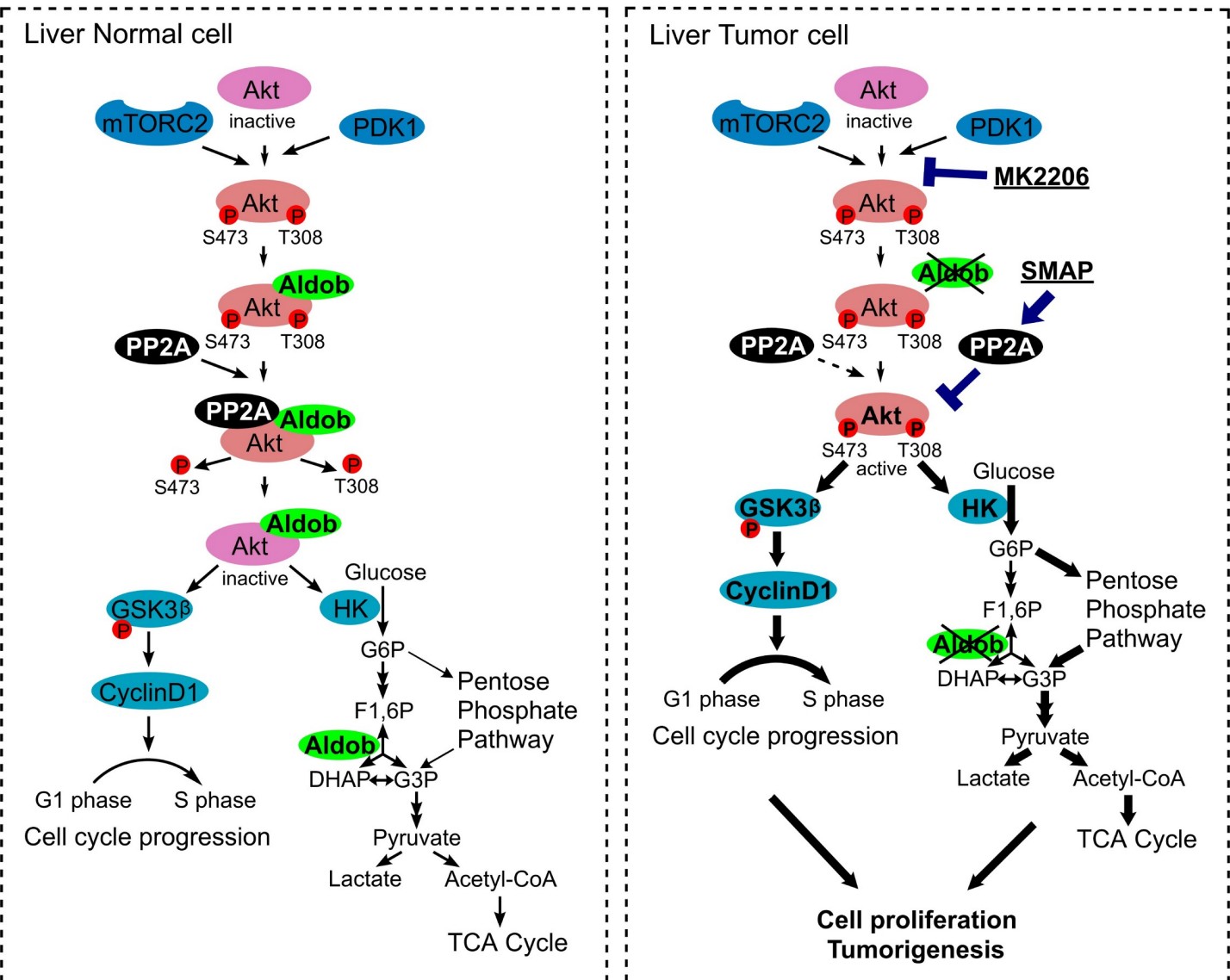

**Fig 8. A schematic model of Aldob-mediated negative regulation of Akt activation.** In normal cells, Aldob interacts with Akt in an Akt kinase activity-dependent manner, which promotes phosphatase PP2A binding to p-Akt and results in p-Akt dephosphorylation. Aldob is involved in the intracellular homeostasis of Akt phosphorylation through modulating substrate/phosphatase interactions. In Aldob-deficient cancer cells, down-regulation of Aldob releases its negative regulation of Akt activity, leading to constitutive activation of oncogenic Akt signaling, which facilitates cell cycle progression and increases glucose consumption and metabolic flux to glycolysis and TCA cycle for cancer cell proliferation and tumorigenesis. Targeting hyperactive Akt through directly inhibiting Akt activity or reactivating PP2A phosphatase activity may be a viable therapeutic approach to treat Aldob-deficient human cancers. Bold arrows represent up-regulation of indicated signaling events in response to the absence of Aldob. Aldob, aldolase B; DHAP, dihydroxyacetone phosphate; GSK3ß, glycogen synthase kinase 3ß; HK, hexokinase; mTORC2, mechanistic target of rapamycin complex 2; p-Akt, phosphorylated Akt; PDK1, phosphoinositide-dependent kinase 1; PP2A, protein phosphatase 2A; SMAP, small-molecule activator of PP2A; TCA cycle, tricarboxylic acid cycle.

alterations in PI3K/Akt pathway, emerging studies have focused on the novel protein interactions with Akt pathway members involved in the regulation of Akt kinase activity [45]. Here, we found that the glycolytic enzyme Aldob directly interacted with Akt resulting in reduced Akt phosphorylation. Although the kinase activity of Akt is critical for its interaction with Aldob, the enzymatic activity of Aldob plays a less important role in this regard. Interestingly, among these 7 single mutants of Aldob, the mutant R304A significantly destroyed this

interaction with Akt, leading to the restoration of Akt phosphorylation and cell growth. The active site residue R304 plays a crucial role in aldolase enzymatic mechanism and inactivating mutant at R304 residue causes HFI [46]. Although there is currently no human data linking HFI to HCC, our results warrant future study on the incidence of HCC in HFI populations, especially those with this mutation. On the other hand, 3 Akt isoforms (Akt1-3) exhibit over 80% identity at amino acid sequence, but are not functionally redundant based on different phenotypes observed by genetic deletion of each isoform in mice [3]. Akt1 played a crucial role in cell survival [47], while Akt2 maintained glucose homeostasis [48]. We found that *ALDOB* KO significantly increased Akt2 phosphorylation at S474 site, and Aldob interacted with Akt2 in HCC cells, suggesting that Aldob plays a similar role in regulating both Akt1 and Akt2 activity. Future studies will examine the role of Aldob and its modulation of Akt isoform-specific signaling networks and biological outcomes.

Emerging studies have unveiled a pleiotropic role of aldolase in diverse physiological and pathological processes, in addition to the well-established role of glycolytic enzyme [12]. Our current work clearly demonstrates the great significance of Aldob-induced repression of Akt activity for the tumor-suppressive function of Aldob in HCC. Moreover, the inverse expression patterns of Aldob and p-Akt suggest that Aldob may be applied as a potential biomarker for HCC treatment with targeted Akt therapeutics. Our previous study identified a novel mechanism by which Aldob interacts with G6PD to reprogram metabolism in HCC [18]. Here, we also highlight the role of Aldob in regulating glucose metabolism through inhibiting Akt activity. Extensive evidence indicates that activated Akt promotes glucose uptake and glycolysis through activating transcription factors and up-regulating many glycolytic enzymes, such as hexokinases [19]. Hexokinases catalyze the ATP-dependent phosphorylation of glucose to glucose-6-phosphate in the first irreversible step of glycolysis and play a critical role in cellular glucose uptake and utilization [49]. HK4, or glucokinase (GCK), is a low-affinity hexokinase with a high Km and is predominantly expressed in the liver and pancreas, but inhibited in HCC cells [50]. However, HK1-2 are high-affinity hexokinases with low Km. HK1 is widely expressed in adult tissues and is considered the housekeeping isoform of hexokinase, and HK2 is highly expressed in many cancer cells [51]. Akt increases the expression and activity of hexokinases by activating HIF1α through stimulation of mTORC1 [52,53]. Down-regulation of hepatic HK1 and HK2 leads to the impairment of glucose uptake and metabolism and HCC cell proliferation [21,54,55]. Indeed, our studies showed a prominent elevation of entire central carbon metabolism in liver tumor tissues of *ALDOB* KO mice compared to WT mice, including glycolysis, TCA, and PPP [18], which was accompanied by activated Akt and increased expression of HK1 and HK2, suggesting that Aldob deficiency facilitates glucose uptake and metabolic flux through up-regulation of hexokinases to sustain tumor growth. Moreover, Aldob-induced inhibition of Akt activity not only reduced glucose uptake and metabolic flux to glycolysis and TCA cycle, but also induced cell cycle arrest through down-regulating Akt/GSK3β/CyclinD1 signaling, thereby leading to the impairment of cell proliferation. Thus, these findings fully demonstrate the important tumor-suppressive role of Aldob through modulating Akt activity and its downstream signaling events.

The mechanistic target of rapamycin (mTOR) is a conserved serine/threonine protein kinase that forms 2 cellular complexes known as mTORC1 and mTORC2, with distinct subunit composition and substrate selectivity [4]. mTORC1 functions as a downstream effector for multiple oncogenic signaling pathways, including PI3K/Akt signaling and Ras/Raf/MEK/ERK (MAPK) signaling. In response to growth factors stimulation, activated Akt phosphorylates the tuberous sclerosis complex 2 (TSC2) and inhibits the GTPase-activating protein (GAP) activity of the tumor suppressor TSC2/TSC1 complex. TSC2 acts as a GAP specific for the GTPase Rheb, an essential positive regulator of mTORC1. Thus, Akt acts as an upstream

activator of mTORC1 [56]. Similarly, ERK also activates mTORC1 via phosphorylation and inhibition of TSC2 [57]. mTORC1 activation promotes the biosynthetic processes of protein, lipid, and nucleotide to support cell growth through phosphorylating numerous downstream substrates such as S6K, while it suppresses the catabolic pathway of autophagy. Our work reveals that loss of Aldob enhances mTORC1 kinase activity due to the activation of mTORC1 upstream activator Akt, but not ERK (Figs 7D and S1B). Ongoing studies will test the effect of Aldob on lipid synthesis and hepatocarcinogenesis. On the other hand, mTORC2 regulates cell proliferation and survival primarily by phosphorylating several members of the AGC (PKA/PKG/PKC) family of protein kinases, such as Akt and PKCα [56]. Interestingly, loss of Aldob has no effect on Akt upstream kinase mTORC2 activity (S1B Fig). Although we have revealed a new tumor-suppressive mechanism of Aldob in HCC, further efforts will be needed to explore potential participation of additional tumor-suppressive mechanism(s) found in other aberrant aldolase isoforms, including Aldoa.

Our current study has identified a close functional relationship among Aldob, Akt, and PP2A, providing new mechanistic insights into the crucial upstream regulators involved in the Akt phosphorylation homeostasis mediated by the coordinate regulation by protein kinases and phosphatases. PP2A is a large family of serine/threonine phosphatases, predominantly found in cellular systems in the form of heterotrimeric holoenzyme consisting of a variable regulatory subunit (B) and dimeric core enzyme formed by a structural subunit (A) and a catalytic subunit (C) [58]. The type of B subunit bound to the core dimer determines both the substrate specificity and cellular localization of PP2A holoenzyme complexes [59]. As a tumor suppressor, PP2A negatively regulates several oncogenic signaling pathways and controls various cellular functions, such as cell growth, cell cycle, and apoptosis [60]. PP2A-B55α has been shown to dephosphorylate Akt through a direct association and negatively regulate Akt-induced cell proliferation and survival [8]. Herein, we demonstrated that Aldob was involved in PP2A-induced inhibition of Akt phosphorylation at both T308 and S473 sites, and PP2A-C plays an important role in Aldob-dependent targeting of the PP2A to Akt. Indeed, we showed that down-regulated Aldob attenuated PP2A interaction and dephosphorylation of Akt, which was consistent with the inverse relationship between Aldob and p-Akt expression in tumor tissues of human HCC and *ALDOB* KO mice. Furthermore, inhibition of PP2A restored Akt phosphorylation in Aldob-expressing cells, while PP2A reactivation using SMAP enhanced Aldob-induced inhibition of Akt phosphorylation. Moreover, alteration of Aldob expression had no effect on PP2A-C expression, and endogenous PP2A-C was identified in Aldob immunocomplex. It is conceivable that Aldob interacts with p-Akt and functions as a scaffold protein to further recruit the serine/threonine phosphatase PP2A to its direct substrate Akt resulting in the inhibition of downstream Akt signaling. Furthermore, the recruitment of PP2A could be mediated through its catalytic component but also through a specific regulatory subunit. It is worthwhile to detect the presence of other B regulatory subunit in the Aldob/Akt/PP2A protein complex in future studies.

Targeting key components of PI3K/Akt pathway is being explored as a therapeutic approach for cancer treatment. Several Akt small molecular inhibitors have entered different stages of clinical trials [3]. MK2206 is a potent and orally bioavailable allosteric inhibitor of Akt that specifically targets Akt inactive conformation and blocks PDK1-mediated phosphorylation and activation [22]. To date, MK2206 has shown moderate antitumor efficacy in vitro and in early clinical studies as a monotherapy and exerts significantly more promising tumor inhibitory activities in combination with other agents, such as HCC first-line treatment sorafenib [61,62]. Our recent study identified a natural polyphenolic compound, term proanthocyanidin B2, extracted from peanut skin, exerts potent antitumor efficacy for HCC through directly targeting and inhibiting Akt enzymatic activity [63]. In this study, we demonstrated

that MK2206 treatment potently inhibited Akt phosphorylation and tumorigenic effects resulted from low Aldob level. Furthermore, tumor cells with low Aldob level were more responsive to Akt inhibition than those with high Aldob expression in which Akt signaling at baseline was already reduced. Collectively, our work suggests that targeting hyperactive Akt due to the loss of Aldob in HCC may be a viable therapeutic strategy.

In addition to direct inhibition of oncogenic kinases, stimulation of endogenous phosphatases to indirectly inactivate kinase signaling has been considered as a promising anticancer therapeutic approach. PP2A is functionally inactivated in many cancers, as a result of various mechanisms including somatic mutation, increased expression of endogenous PP2A inhibitors, and posttranslational modifications of the catalytic subunit [40]. Aberrant transcripts of tumor suppressor gene *PPP2R1B* were detected in 29% of HCC tumors, which associated with the development of HCC [64]. Overall, the activation of PP2A has the potential to exert antitumor effect toward multiple oncogenic signaling proteins that drive cancer progression. FTY720 (fingolimod) has been reported to indirectly activate PP2A through inhibiting the endogenous PP2A inhibitor SET and exhibits high antitumor efficacy against HCC [39,65]. In this study, we report that an orally bioavailable PP2A activator SMAP has anti-HCC activity in vitro and in vivo through its ability to simultaneously inhibit PP2A substrates Akt and ERK activity, resulting in cell cycle arrest and ultimately cell death. SMAP mediated PP2A activation potentiated Aldob-induced suppression of Akt signaling and tumor growth. Our findings provide evidence that the restoration of PP2A tumor suppressive phosphatase activity could represent a novel targeted therapeutic approach for HCC treatment. Future studies are needed to validate the therapeutic effects of PP2A reactivation in Aldob-defective cancers.

In summary, our study uncovers the biological importance of Aldob-mediated Akt inhibition through the formation of Aldob/Akt/PP2A protein complex, independent of Aldob enzymatic activity. Our study suggests that targeting hyperactive Akt resulting from Aldob deficiency through directly inhibiting Akt kinase or reactivating PP2A phosphatase activity may serve as an antitumor treatment for HCC.

## Methods

### Ethics statement

This study was approved by the Clinical Research Ethics Committee of the Eastern Hepatobiliary Surgery Hospital, Shanghai, China. Written informed consent was obtained from all patients. All specimens were pathologically confirmed at the Eastern Hepatobiliary Surgery Hospital. Animal studies were approved by the Institutional Animal Care and Use Committee of the Shanghai Institutes for Biological Sciences of the Chinese Academy of Sciences (approval number 2015-AN-2).

### Mouse models

Mice were housed in pathogen-free animal care facilities in Shanghai Institutes for Biological Sciences (SIBS), Chinese Academy Science (CAS) and allowed access to water and chow diet ad libitum. All animal experimental procedures were conformed to SIBS Guide for the Care and Use of Laboratory Animals and were approved by the Institutional Animal Care and Use Committee of SIBS, CAS. For subcutaneous xenograft growth model, Huh7-Aldob or Vector cells ($2 \times 10^6$) mixed in serum-free medium containing 50% Matrigel (BD Biosciences, San Jose, California, United States of America) were subcutaneously injected into the left and right flanks of 5-week-old male BALB/C nude mice. About 10 days after cell injection, when the tumors were visible and reached 100 mm$^3$, the mice were randomly assigned to control solvent and agent treatment groups ($n = 6$ per group). MK2206 (40 mg/kg, once every other day) and

SMAP (5 mg/kg, twice a day) were administered via oral gavage. Tumor growth and body weight were monitored every other day for 3 weeks. Tumor volume was estimated using the following formula: volume = $0.52 \times$ length $\times$ width$^2$. Animals were observed for signs of toxicity (weight loss, abdominal stiffness, and ruffled fur). Tumors were harvested, photographed, and weighted 2 hours after the final dose of treatment, following snap-frozen in liquid nitrogen for immunoblotting or fixation in 4% paraformaldehyde (PFA) for IHC. IHC staining was performed to examine the protein expression of proliferation marker gene Ki67 in the tumor tissues.

The whole-body *ALDOB* KO mouse model and the liver-specific *ALDOB* KO (ALDOB$^{f/f}$; *Alb*-Cre) mouse model were generated as previously described [18]. *ALDOB* KO and WT mice were injected with a single dose of DEN (40 mg per kg body weight) at postnatal day 14. These mice were sacrificed at the age of 40 weeks, and mouse liver, tumor tissues, and plasma were collected.

## Cell culture and HCC tissue samples

The human HCC cell lines Huh7 and HCC-LM3 were purchased from Cell Bank of CAS. Primary hepatocytes were isolated from the livers of male liver-specific *ALDOB* KO and WT mice using collagenase perfusion as previously described [66]. HCC cell lines and primary hepatocytes were cultured in Dulbecco's Modified Eagle's Medium (DMEM, Hyclone, Logan, Utah, USA) plus with 10% fetal bovine serum (FBS, Hyclone) and 1% penicillin-streptomycin (Gibco, Carlsbad, California, USA) in a 37˚C incubator with 5% $CO_2$ in air.

Human HCC tissues were collected immediately after hepatectomy from the Eastern Hepatobiliary Surgery Hospital, Shanghai. Collection of patient specimens with informed consent to an established protocol and all experimental procedures were approved by the Research Ethics Committee of Eastern Hepatobiliary Surgery Hospital. The fresh specimens were snap-frozen in liquid nitrogen and stored at −80˚C until analysis.

## Reagents

MK2206 (Selleck Chemicals, Houston, Texas, USA) was dissolved in DMSO (Sigma-Aldrich, St. Louis, Missouri, USA) and diluted in DMEM with a final concentration of 0.1% DMSO for in vitro studies or in the vehicle solution containing Cremophor (Sigma-Aldrich), 95% ethanol and water in a ratio of 1:1:6 for in vivo studies. EGF (Sigma), insulin (Invitrogen), and OA (Cell Signaling Technology, Boston, Massachusetts, USA) were used at the indicated doses. The PP2A activator SMAP compound (provided by Goutham Narla, The University of Michigan, Ann Arbor, Michigan) was dissolved in DMSO and stored at room temperature and used for up to 1 month for in vitro studies. For in vivo studies, SMAP was prepared in an N, N-Dimethylacetamide (DMA)/Kolliphor HS-15 (Solutol)/diH$_2$O solution.

## Plasmids and viruses

Aldob, Akt1, and Akt2 expression plasmids were constructed by cloning the open reading frame of each cDNA into the multiple cloning site of pCDNA3.0 vector, and their different Flag or Myc tag was constructed through design in 1 primer. The different gene fragments were amplified using their primers. The pcDNA3.0 vector and each PCR fragments were both digested with their restriction enzymes EcoRI and XhoI or BamHI and XhoI or EcoRI and XhoI, and then ligated with T4 ligase. Transformation was performed using DH5α competent cells. Single clones were selected and amplified to extract plasmids. All clones were sequenced to confirm the results. Flag-PP2A-C plasmids were obtained from Dr. Xie Dong (SIBS, CAS). Plasmid transfection and virus infection: Expression constructs were transfected into cells

using Lipofectamine 2000 Transfection Reagent (Invitrogen) following the manufacturer's protocol. Lentivirus expression clone pLVX-IRES-GFP-puromycin-Aldob with/without Flag-tag was constructed with the Aldob fragment digested from pcDNA3.0-Aldob mentioned above via restriction enzyme EcoRI and XhoI. This fragment was inserted into pLVX-IR-ES-GFP-Puro vector and sequenced to verify the results. Lentivirus was produced by co-transfection of 293T cells using Lipofectamine 2000 Transfection Reagent (Invitrogen) with 3 μg pCMV-dR8.91, 2 μg pCMV-VSV-G, and the target gene expression vectors (such as 5 μg pLVX-Aldob-Flag-puro). Fresh medium was transferred into the dish after transfection of 6 hours. After 48 hours, the virus was harvested from the medium filtered by 0.45 mm Steriflip-filter (Millipore, Boston, Massachusetts, USA). Virus infection was carried out by incubating cells with cell medium containing indicated virus and 5 μg/ml polybrene (Sigma) for 8 to 12 hours. Cells were allowed to recover in complete medium for 24 hours and then selected with antibiotic (such as puromycin) for 10 to 15 days until stable cell lines were screened and constructed.

## RNA-mediated interference (RNAi)

siRNA-Aldob-2-F: 5′-ACCCUCUACCAGAAGGACAdTdT-3′, siRNA-Aldob-2-R: 5′-UGUC CUUCUGGUAGAGGGUdTdT-3′, siRNA-Aldob-3-F: 5′-GUAUGUUCACACGGGUUCU dTdT-3′, siRNA-Aldob-3-R: 5′-AGAACCCGUGUGAACAUACdTdT-3′. siRNA-Akt1-F: 5′-GCUACUUCCUCCUCAAGAAdTdT-3′, siRNA-Akt1-R: 5′-UUCUUGAGGAGGAAGUAG CdTdT-3′. The esiRNA targeting human ALDOB (EHU001201) was obtained from Sigma-Aldrich. The siRNA was transfected into cells using Lipofectamine 2000 Transfection Reagent (Invitrogen) for at least 48 hours and then subjected to different assays.

## Western blotting and antibodies

Cells or tissues were lysed in ice-cold lysis buffer (25 mM Tris (pH 8.0), 150 mM NaCl, 1 mM CaCl$_2$, 1% Triton X-100) with EDTA-free protease inhibitors (Bimake, Houston, Texas, USA) and phosphatase inhibitors (Bimake) and determined the protein concentration though BCA protein assay kit (Beyotime Biotechnology, Shanghai, China). Equal amounts of total protein were subjected to 10% sodium dodecyl sulfate (SDS)-polyacrylamide gel electrophoresis (PAGE) and then transferred to polyvinylidene fluoride (PVDF, Millipore) membranes and incubated with the indicated antibodies. Antibodies to Aldob (18065-1-AP), GAPDH (60004-1-Ig), Tubulin (11224-1-AP), p70S6K (14485-1-AP), PHLPP (22789-1-AP), p21 (10355-1-AP), p27 (25614-1-AP), PDK1 (17086-1-AP), PP2A-C subunit (13482-1-AP), GST-tag (66001-2-AP), Hexokinase1 (19662-1-AP), Hexokinase2 (22029-1-AP), and Hexokinase4 (19666-1-AP) were purchased from Proteintech (Rosemont, USA). Antibodies to pT308-Akt (13038), pS473-Akt (4060), Akt1 (2938), pS9-GSK3β (5558), GSK3β (12456), pT389-S6K (9205), Myc-tag (2276), IgG (2729), pT202/Y204-ERK (4370), ERK (4695), pT638/641-PKCα (9375), PP2A-A subunit (2041), PP2A-B subunit (2290), and p70S6K (2708) were purchased from Cell Signaling Technology. Antibodies to Flag-tag (F3165) were purchased from Sigma. Antibodies against PKCα (sc-17769) were obtained from Santa Cruz. Antibodies to pT308-Akt (ab38449, for IHC use), CyclinD1 (ab16663), and Ki67 (ab15580, for IHC use) were purchased from Abcam (Cambridge, United Kingdom). After extensive rinse with TBST, immunoreactive bands were incubated with horseradish peroxidase (HRP)-conjugated secondary antibodies and visualized using a chemiluminescence (ECL) kit (Proteintech) through Tanon 5200 Chemiluminescent Imaging System. All western blots were analyzed with ImageJ for quantitative measurements.

## Cell viability assay

The viability assay was performed by seeding cells in a 96-well plate at a density of 2,000 cells/well for 6 repeats. After 7 hours, cell media were replaced with fresh media with various concentrations of test agents or control DMSO, and cells were proceeded to grow. CCK8 (OBIO Cell Counting Kit) then was added to separate plates at the indicated time according to the manufacturer's instruction.

## Colony formation assay

Colony formation assays were adapted from a protocol reported previously [34]. Cells of different groups were seeded (5,000 cells/well) into 6-well plates and cultured for 7 to 10 days. Medium were changed every other day. Colonies were washed with PBS, fixed with 10% acetic acid/10% methanol for 20 minutes, and then stained with 0.4% crystal violet in 20% ethanol for 20 minutes. After staining, the plates were washed and air-dried for colony visualization.

## Cell cycle analysis

Cells of different groups were plated ($2 \times 10^5$ cells/well) into 6-well plates and subjected to indicated treatment. Resulting cells were detached with trypsin, washed with ice cold PBS, and fixed in 70% ethanol overnight. About $1 \times 10^6$ cells were detected using the cell cycle analysis kit (Beyotime Biotechnology, C1052). Then, cells were incubated with propidium iodide (PI) and RNAase for 30 minutes at 37°C, followed by analysis via the flow cytometry (BD Bioscience). The results were presented as the proportion of cell population in each phase of the cell cycle, which was determined using FlowJo 7.6 software (Treestar, USA).

## Measurement of glucose consumption and lactate production

Cells were seeded and cultured with the medium containing 2 g/l glucose. The culture medium was collected at indicated time points. The D-Glucose Kit (#10716251035, R-Biopharm) and the L-Lactic Acid Kit (#10139084035, R-Biopharm) were used to detect the concentration of glucose and lactate in the culture medium and plasma according to the manufacturer's instructions.

## Aldolase activity assay

Aldolase catalyzes the reversible reaction of fructose-1,6-bisphosphate (F-1,6-BP) to glyceraldehyde-3-phosphate and dihydroxyacetone phosphate. The measurement of Aldolase activity was performed using F-1,6-BP as the substrate by Aldolase Activity Colorimetric Assay Kit (#K665-100; BioVision, San Francisco, USA) according to the manufacturer's instructions.

## Metabolic flux experiments using [U-$^{13}$C$_6$]-glucose and Gas chromatography/mass spectrometry

The metabolic flux analysis using [U-$^{13}$C$_6$]-glucose followed our previously published protocol [18,63,67]. In short, cells of different groups were seeded at a density of approximately $2 \times 10^6$ cells per 10-cm dish. After cells grew to approximately 60% confluence, the unlabeled medium was replaced with [U-$^{13}$C$_6$]-glucose medium, which composed of low glucose DMEM (Gibco, 11054–020: 1 g/L unlabeled glucose, no glutamine) with supplement of 1 g/L [U-$^{13}$C$_6$]-glucose (Cambridge Isotope Laboratory), 10% (v/v) fetal bovine serum, 1 mM pyruvate, 2 mM unlabeled L-glutamine, and 1% (v/v) penicillin-streptomycin with or without MK2206 treatment (5 μM) for 12 hours.

## Immunoprecipitation (IP) and GST pull-down assay

A total of 1,000 μg cell lysates were incubated with the indicated antibody (1 to 2 μg) overnight at 4˚C with gentle rotation, followed by 6-hour incubation with Protein A/G Sepharose beads (Santa Cruz). Then, beads were washed 4 times with IP lysis buffer (150 mM Tris-HCl, 400 mM NaCl, 0.8%Triton X-100 (pH 7.4)) and boiled in sample buffer before being resolved by SDS-PAGE and immunoblotted with indicated antibodies. For the Flag or Myc-tag labeled IP experiments, cell lysates were incubated with antibody-Flag or antibody-Myc affinity gel (Bimake) overnight at 4˚C. Affinity gel were collected after 300*g* centrifuging for 4 minutes at 4˚C, and then washed 4 times. The subsequent operations were similar to western blot analysis. For GST pull-down assay, recombinant GST-Akt1 or GST proteins were incubated with His-Aldob proteins or cell lysates derived from cells transfected with indicated constructs in binding buffer (25 mM Tris-HCl, 200 mM NaCl, 1 mM EDTA, 0.5% NP-40, 10 μg/μl BSA, 1 mM DTT (pH 7.3)). A total of 100 μl prepared Glutathione Sepharose 4B (GE Healthcare, Little Chalfont, Buckinghamshire, UK) were added to the mixture and incubated overnight. After washed in ice-cold binding buffer for 3 times, bound proteins were released by boiling in sample buffer and resolved by SDS-PAGE electrophoresis.

## Expression and purification of recombinant proteins

The full-length Aldob and Akt1 genes were inserted into the pET-30a (Novagen, Madison, Wisconsin, USA) and pGEX4T1 (GE Healthcare) expression plasmids, respectively. The *E. coli* BL21 (DE3) Codon-Plus strain (Novagen) transformed with the reconstructed expression plasmid was cultured at 37˚C in LB medium containing 0.05 mg/ml ampicillin until OD600 of medium reached 0.8, and addition of 0.1 mM IPTG induced the target protein expression at 16˚C for 24 hours. The cells were further harvested by centrifugation and lysed by sonication in the lysis buffer (20 mM Tris-HCl, 300 mM NaCl, and 1 mM PMSF). The recombinant proteins were purified using GSH Magnetic Beads for GST tag protein purification (Bimake, B23702) and Nickel Magnetic Beads for His tag protein purification (Bimake, B23602) according to the manufacturer's protocol. The purified proteins were determined by SDS-PAGE with high purity (>95%).

## In vitro Akt kinase assay

The in vitro kinase assay (Abcam, ab65786) was performed according to the manufacturer's instructions. Cell lysates were immunoprecipitated with either anti-Akt1 or anti-IgG antibody along with protein A/G beads overnight, and the immunoprecipitates were incubated with substrate GSK3α Protein/ATP mixture in kinase reaction buffer for 30 minutes at 30˚C. The reactions were terminated by 6×SDS loading buffer and subjected to western blot analysis with antibody against pS21-GSK3α.

## In vitro PP2A dephosphorylation assay

The in vitro PP2A dephosphorylation assay was performed using phosphorylated Myc-Akt1 as the substrate. Myc-Akt1 transfected Huh7 cells were serum-starved for 24 hours and then stimulated with 0.1 μM insulin for 1 hour before harvesting. Equal amounts of cell lysates were incubated with antibody-Myc affinity gel overnight at 4˚ C. The immunoprecipitates were washed 3 times in IP lysis buffer, 3 times in PP2A phosphatase assay buffer (20 mM HEPES (pH 7.2), 100 mM NaCl, and 3 mM DTT) and finally resuspended in PP2A phosphatase assay buffer. Then, immunoprecipitated phospho-Akt were reacted with recombinant active PP2A (BPS Bioscience, 30056) at 30˚C for indicated time periods in the presence or absence of

recombinant WT-Aldob or Aldob-R304A mutant. Dephosphorylation reactions were terminated by 6×SDS loading buffer and subjected to western blot analysis.

## Tissue microarray and immunohistochemistry

IHC was constructed on the TMA with a 2-step immunoperoxidase technique [68]. In short, tissues were fixed overnight in 4% PFA, then dehydrated in ethanol and embedded in paraffin. The paraffin-embedded tissue pieces were cut into 5-μm thick sections, which were stained with hematoxylin and eosin (H&E) and processed for IHC. For IHC, specimen sections were dewaxed, rehydrated in ethanol, and subjected to antigen extraction. Endogenous peroxidase activity was blocked with 0.3% hydrogen peroxide in methanol for 30 minutes. Sections were blocked by 2% BSA in PBS for 1 hour at 37˚C, and then incubated with the indicated primary antibody overnight at 4˚C. After washing 3 times with PBS the next day, the samples were further incubated for 1 hour at 37˚C using the corresponding secondary antibodies. The slides were displayed by DAB staining. Reading of IHC slides and TMA was performed using Vectra 2 (Perkin Elmer, Waltham, Massachusetts, USA) to quantify protein expression in all samples. For TMA in Fig 1B, A Leica APERIO AT TURBO instrument was used for scanning and analysis. Protein expression intensity was stratified into negative (0 to 20 score), weak (21 to 100 score), moderate (101 to 180 score), and strong (181 to 255 score). The calculation formula was (weak + moderate + strong)/(negative + weak + moderate + strong) × 100%. Quantification of IHC staining was performed blindly. TMA antibody characterization of Aldob and pT308-Akt was shown in S1A Fig. For tissue IHC staining of Ki67 in S6A and S8E Fig, Ki67 IHC score was estimated using the following formula: IHC score = intensity × area. The intensity was graded as 0, negative; 1, weak; 2, moderate; and 3, strong. The area of positive staining was assessed by 0, 1% to 25%, 26% to 50%, 51% to 75%, and 76% to 100%, corresponding to 0, 1, 2, 3, and 4 grades, respectively.

## Statistical analysis

All data are presented as the means ± standard errors of the mean from at least 3 independent experiments. The survival curve was calculated according to the Kaplan–Meier method, and differences were assessed by the log-rank test. Differences between measurable variants of 2 groups were calculated by the unpaired Student $t$ test or 2-way ANOVA using GraphPad Prism 5.0, and the value of $p < 0.05$ was considered statistically significant.

## Supporting information

**S1 Fig. Aldob deficiency enhances Akt and mTORC1, but not mTORC2, kinase activity in diethyl nitrosamine-induced HCC mouse models. (A)** TMA antibody characterization of Aldob and pT308-Akt (original magnification ×200). **(B)** IB analysis of WCL derived from liver tumor tissues of WT and *ALDOB* KO mice after injection with DEN at postnatal day 14 to induce hepatocellular carcinoma for 10 months. The data underlying this figure can be found in S2 Data. Aldob, aldolase B; DEN, diethyl nitrosamine; HCC, hepatocellular carcinoma; IB, immunoblot; KO, knockout; mTORC1, mechanistic target of rapamycin complex 1; TMA, tissue microarray; mTORC2, mechanistic target of rapamycin complex 2; WCL, whole cell lysate.
(TIF)

**S2 Fig. Forced expression of the activated form of Akt, but not the Akt phospho-deficient mutant, in Aldob-overexpressing cells eliminates Aldob-mediated tumor-suppressive effects. (A)** Cell viability assay of Huh7-Vector and Huh7-Aldob cells transfected with Myc-

Ctrl or Myc-Akt1 constructs. **(B)** Representative graphs from colony formation assay of Huh7-Vector and Huh7-Aldob cells after transfection with Myc-Ctrl or Myc-Akt1 for 7 days. **(C and D)** Cells were transfected with Myc-Ctrl or Myc-Akt1 for 48 hours, and then monitored for cell cycle distribution **(C)** or subjected to IB analysis **(D)**. **(E)** Cell viability assay of Huh7-Vector and Huh7-Aldob cells transfected with the indicated constructs. Myc-Akt1-AA indicated the Myc-Akt1 phospho-deficient mutant harboring duple mutations (T308A/S473A). **(F)** IB analysis of WCL derived from Huh7 cells transfected with the indicated constructs. Data are presented as mean ± SEM. $^*$ $p < 0.05$; $^{**}$ $p < 0.01$ (Student $t$ test). The data underlying this figure can be found in S2 Data. Aldob, aldolase B; IB, immunoblot; WCL, whole cell lysate.
(TIF)

**S3 Fig. Akt inhibition through MK2206 suppresses the oncogenic effects mediated by knockdown of Aldob. (A)** Cell viability of Huh7 cells transfected with indicated siRNAs in the presence of DMSO or MK2206 (2 μM). **(B and C)** Glucose consumption **(B)** and lactate production **(C)** of Huh7 cells transfected with indicated siRNAs after treatment with DMSO or MK2206 (5 μM). **(D)** IB analysis of WCL derived from Huh7 cells transfected with indicated siRNAs in the presence of DMSO or MK2206 (2 μM). Data are presented as mean ± SEM. $^*$ $p < 0.05$; $^{**}$ $p < 0.01$ (Student $t$ test). The data underlying this figure can be found in S2 Data. Aldob, aldolase B; IB, immunoblot; siRNA, small interfering RNA; WCL, whole cell lysate.
(TIF)

**S4 Fig. Inhibition of Akt kinase activity is essential for Aldob-induced antitumor effects. (A–D)** Aldob-overexpressing Huh7 cells transfected with indicated siRNAs were used to determine their effects with or without MK2206 treatment (2 μM) on cell proliferation **(A)**, colony formation **(B)**, cell cycle distribution **(C)**, and cell cycle–related protein levels **(D)**. **(E and F)** Glucose levels in the culture medium of Huh7-Aldob cells transfected with indicated siRNAs after treatment with DMSO or MK2206 (5 μM) at different time points. **(G and H)** Fraction of the labeled metabolites of M+3 from $^{13}$C-glucose in glycolysis by DMSO or MK2206 (5 μM) treatment for 12 hours in Huh7-Aldob cells transfected with indicated siRNAs. **(I and J)** Fraction of the labeled metabolites of M+2 from $^{13}$C-glucose in TCA cycle by DMSO or MK2206 (5 μM) treatment for 12 hours in Huh7-Aldob cells transfected with indicated siRNAs. Data are presented as mean ± SEM. $^*$ $p < 0.05$; $^{**}$ $p < 0.01$ (Student $t$ test). The data underlying this figure can be found in S2 Data. Aldob, aldolase B; siRNA, small interfering RNA; TCA, tricarboxylic acid.
(TIF)

**S5 Fig. Aldob inhibits HCC cell growth through suppression of Akt signaling. (A–D)** LM3 cells stably expressing Aldob via lentiviral infection (with Vector as a negative control) were used to examine their biological functions in the presence of either control DMSO or MK2206 (2 μM), including cell proliferation **(A)**, colony formation **(B)**, cell cycle distribution **(C)**, and the protein levels of Akt pathway **(D)**. **(E)** Co-IP analysis to demonstrate the interaction between exogenous Aldob and endogenous Akt1 in LM3-Aldob cells. Data are presented as mean ± SEM. $^*$ $p < 0.05$; $^{**}$ $p < 0.01$ (Student $t$ test). The data underlying this figure can be found in S2 Data. Aldob, aldolase B; HCC, hepatocellular carcinoma; IP, immunoprecipitation.
(TIF)

**S6 Fig. Aldob suppresses tumor growth in vivo through inhibition of Akt signaling. (A)** Representative IHC images and quantification of Ki67 expression in Huh7-Vector and Huh7-Aldob xenograft tumors treated with control solvent or MK2206 ($n = 6$). Scale bars,

50 μm. **(B)** IB analysis of WCL derived from Huh7-Vector and Huh7-Aldob tumors treated with control solvent or MK2206. **(C)** The body weights of mice in Fig 3I were recorded. Data are presented as mean ± SEM. $^*$ $p < 0.05$; $^{**}$ $p < 0.01$ (Student $t$ test). The data underlying this figure can be found in S2 Data. Aldob, aldolase B; IHC, immunohistochemistry; WCL, whole cell lysate.
(TIF)

**S7 Fig. Aldob directly interacts with Akt to promote PP2A-C binding to Akt and inhibit Akt activity. (A)** Co-IP assay to show that Akt2 interacted with Aldob in Huh7 cells at ectopic expression conditions. **(B)** IP analysis was performed with WCL derived from Myc-Akt1 transfected Huh7 cells and various truncated mutants of recombinant His-Aldob proteins to illustrate that the carboxyl-terminal region (a.a. 241–364) of Aldob is responsible for Akt1-binding. **(C** and **D)** IB analysis of Flag-IP and WCL derived from Huh7 cells transfected with the indicated constructs. The data underlying this figure can be found in S2 Data. Aldob, aldolase B; IP, immunoprecipitation; PP2A-C, the catalytic subunit of protein phosphatase 2A; WCL, whole cell lysate.
(TIF)

**S8 Fig. PP2A activation with SMAP reduces Akt phosphorylation and suppresses HCC cell growth. (A)** IB analysis of WCL derived from Huh7 and LM3 cells treated with SMAP for 24 hours at the indicated concentrations. **(B)** Relative cell viability of Huh7 and LM3 cells treated with increasing concentrations of SMAP for 48 hours. **(C)** The body weights of mice in Fig 7I were recorded. **(D)** IB analysis of WCL derived from Huh7-Vector and Huh7-Aldob tumors treated with control solvent or SMAP. **(E)** Representative IHC images and quantification of Ki67 expression in Huh7-Vector and Huh7-Aldob xenografts treated with control solvent or SMAP ($n = 6$). Scale bars, 50 μm. Data are presented as mean ± SEM. $^*$ $p < 0.05$; $^{**}$ $p < 0.01$ (Student $t$ test). The data underlying this figure can be found in S2 Data. Aldob, aldolase B; HCC, hepatocellular carcinoma; IB, immunoblot; IHC, immunohistochemistry; PP2A, protein phosphatase 2A; SMAP, small-molecule activator of PP2A; WCL, whole cell lysate.
(TIF)

**S1 Table. Association of clinicopathological characteristics with Aldob and p-Akt expression in HCC patients (n = 70).**
(DOCX)

**S1 Data. Numerical data related to the main figures.**
(XLSX)

**S2 Data. Numerical data related to the supplementary figures.**
(XLSX)

**S1 Raw Images. All the original images supporting western blot results.**
(PDF)

## Acknowledgments

We thank Dr. Dong Xie (Shanghai Institute of Nutrition and Health, Chinese Academy of Sciences, Shanghai, China) for critical technical support. We acknowledge the help from molecular biology core laboratory, animal facilities, and mass spectrometry facilities at Shanghai Institutes of Nutrition and Health (SINH), CAS, Shanghai.

## Author Contributions

**Conceptualization:** Xuxiao He, Huiyong Yin.

**Funding acquisition:** Yongzhen Tao, Huiyong Yin.

**Investigation:** Xuxiao He, Min Li, Hongming Yu, Guijun Liu, Ningning Wang, Chunzhao Yin, Qiaochu Tu.

**Methodology:** Xuxiao He, Min Li.

**Project administration:** Huiyong Yin.

**Resources:** Hongming Yu, Goutham Narla, Shuqun Cheng.

**Supervision:** Yongzhen Tao, Shuqun Cheng, Huiyong Yin.

**Validation:** Huiyong Yin.

**Writing – original draft:** Xuxiao He.

**Writing – review & editing:** Goutham Narla, Huiyong Yin.

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
