## [Editor Report · Decision Letter 0]

2 Jun 2020

Dear Dr Yin, 

Thank you for submitting your manuscript entitled "Loss of hepatic aldolase B activates Akt and promotes hepatocellular carcinogenesis by destabilizing Aldob/Akt/PP2A protein complex" for consideration as a Research Article by PLOS Biology.

Your manuscript has now been evaluated by the PLOS Biology editorial staff as well as by an academic editor with relevant expertise and I am writing to let you know that we would like to send your submission out for external peer review.

Please re-submit your manuscript within two working days, i.e. by Jun 04 2020 11:59PM.

Kind regards,

Di Jiang,

Associate Editor

PLOS Biology

---

## [Decision Letter · Decision Letter 1]

8 Jul 2020

Dear Dr Yin,

Thank you very much for submitting your manuscript "Loss of hepatic aldolase B activates Akt and promotes hepatocellular carcinogenesis by destabilizing Aldob/Akt/PP2A protein complex" for consideration as a Research Article at PLOS Biology. Your manuscript has been evaluated by the PLOS Biology editors, an Academic Editor with relevant expertise, and by three independent reviewers.

As you will see, the reviewers find your work interesting, but they also raise several concerns that need to be addressed. Regarding the related Nature Cancer manuscript, while we agree that the results seem to be complementary, as it is currently presented this manuscript is conceptually based on the results described in the other manuscript and you do need to describe carefully how the two works overlap and what is novel on this one. We can see that the other paper was published this week, so this should facilitate the revision. You should also address all the other points, which are mostly important to strengthen the mechanistic aspects and we do think that addressing these points will improve the technical quality of the manuscript.

In light of the reviews (attached below), we will not be able to accept the current version of the manuscript, but we would welcome re-submission of a much-revised version that takes into account the reviewers' comments. We cannot make any decision about publication until we have seen the revised manuscript and your response to the reviewers' comments. Your revised manuscript is also likely to be sent for further evaluation by the reviewers.

We expect to receive your revised manuscript within 2 months. 

**IMPORTANT - SUBMITTING YOUR REVISION**

*Re-submission Checklist*

*Published Peer Review*

*PLOS Data Policy*

*Blot and Gel Data Policy*

Sincerely,

Ines Alvarez-Garcia, PhD

Senior Editor

PLOS Biology

on behalf of

Di Jiang, PhD, 

Senior Editor

PLOS Biology

Reviewers' comments

Rev. 1:

The authors present evidence that loss of the glycolytic enzyme Aldob (the liver specific isoform) leads to a poor prognosis in hepatocellular carcinoma. Mechanistically, Aldob nucleates a complex with the general cellular phosphatase PP2A and Akt to dephosphorylate and inactivate Akt. Thus Aldob loss in tumor tissues allows constitutive Akt signaling and oncogenic growth. They show that pharmacological activation of PP2A attenuates tumor cell proliferation in vitro and in a xenograft model, suggesting a possible therapeutic application.

 The data presented clearly show an inverse correlation between Aldob expression and Akt activity. However, as discussed below, there are issues as to how these findings can be consistent with known role of Akt in cancer and glucose metabolism and some question as to how much of what is currently presented is novel. These concerns will have to be addressed before it can be accepted for publication.

Major points

1) The authors must probe for expression for Aldoa expression. Aldoa is ectopically expressed in many cancers. Although it is the muscle-specific isoform, in fact it becomes the most expressed aldolase isoforms in many cancers (Chang et al, 2018, Trends Endocrinol. Metab. 29:549-559). While the mechanism presented here is independent of Aldob enzymatic activity and Aldoa expression would not affect the main findings of the paper, nevertheless this would have implications on the metabolic aspect and suggest that it is not necessary to divert glucose into the PPP to bypass the aldolase step.

2) The authors make frequent references to a recent publication of theirs and that the current work is an extension of it. However this work is not cited in the references and I could not find it on Pubmed. Without any knowledge of its contents it is not possible to fairly assess the current manuscript as to whether it agrees with the earlier findings, or how much of the work being presented is novel.

3) Akt activation increases glucose uptake via glucose transporters and stimulation of hexokinase activity. The only way to resolve this with the loss of Aldob (assuming Aldoa is not expressed) is that glucose enters the PPP shunt and re-enters the glycolytic pathway as G3P. The authors should measure glucose uptake to see if it is increased or unchanged in their cell or mouse panel. HK1 levels are measured in mouse livers in Fig2A. However liver predominantly utilizes HK4, while HK2 is the isoform that is activated by Akt. Therefore the conclusion that a slight increase in HK1 levels is indicative of increased glucose metabolism in Aldob deficient cells is a very weak one. The authors need to show that glucose is taken up and mobilized to G6P in a way consistent with their model.

 In the same direction, the metabolomic data presented in Fig3 does not address the questions raised. All that is shown is a general reduction in metabolites from glycolysis and the TCA upon Aldob re-expression or Akt inhibition. This reduction is across the board and in any case does not resolve the question as to how to reconcile Akt activation and increased glucose uptake with loss of aldolase activity. The authors should re-analyse their data to look for an increase in PPP intermediates to see if glucose is indeed diverted into the PPP.

Minor points

1) Fig 1b. No controls shown for the IHC from the HCC patients. Staining from the adjacent non-malignant tissue should be shown.

Rev. 2: Mónica Álvarez-Fernández - note that this reviewer has waived anonymity

In this manuscript the authors show a novel and direct link between the glycolytic enzyme Fructose-1, 6-biphosphate aldolase B (Aldob) and AKT. Through an extensive set of rigorous and well-perfomed biochemical assays, they show that Aldob directly interact with AKT, preferentially with its phosphorylated and active form, and favors the recruitment of PP2A to the complex, which leads to the inactivation of AKT. They also propose that this new function of Aldob contributes to the tumor suppressive role of this enzyme in HCC, and that interfering with it, either by inhibiting AKT or activating PP2A through SMAP, might have therapeutic value for HCC treatment. The authors also nicely show an inverse correlation between the expression of Aldob and phosphor-Akt in HCC patients, which has prognostic value and supports their model. Overall this is an interesting study, however some points should be addressed before been accepted for publication. 

1) The authors refer several times to their own previous work in which loss of Aldob promotes DEN-induced HCC in an ALDOB KO mouse model. However, there is no reference for this work along the manuscript and I have not been able to find that study across the literature. Can the authors explain that? Are those unpublished data? This should be clarified and we should have access to those data, since this is the basis for this study and a key point for the relevance of the data presented in this manuscript. 

2) Authors conclude that AKT inhibition suppresses the tumorogenesis caused by loss of Aldob. This is not fully demonstrated here since they use an overexpression model for AldoB on AldoB deficient cell lines, in which AKT inhibition or SMAP-mediated PP2A activation mimics the effect of Aldob expression. Authors should limit that conclusions, or perform additional experiments to support that conclusion, that is, a loss of function model, such as the DEN-induced HCC model in the ALDOB KO mouse referred in the manuscript (see above), in which AKT inhibition (and SMAP treatment) should rescue the tumorogenesis induced by Aldob loss according to their model. 

3) PP2A is a multimeric enzyme and form multiple complexes with different regulatory subunits. Therefore, the recruitment of PP2A could be mediated through its catalytic component but also through a specific regulatory subunit. Indeed, one of this B subunit, B55a, has been related to AKT dephosphorylation. All assays shown in the manuscript are based on the catalytic subunit of PP2A. Have the authors evaluated the presence of B55a or any other B regulatory subunit in the complex? At least some comments on the potential specific PP2A complexes involved in this new mechanism of regulation should be added to the discussion.

Minor points:

1. Figure 1: The intensity scoring system used to stratify the expression of pAkt and Aldob is not clear. In methods section two different scoring systems are mentioned, this is confusing, and it is not clear whether the same scoring system was used for both markers. 

2. Figure 2: Details should be provided on the ALDOB KO mouse model. Is it inducible or constitutive? Which promoter uses? This is not referenced in the text nor even mentioned in the methods section. On Fig 2A, for instance, if deletion is performed in vitro upon Cre expression, expression of Cre on ALDOB+/+ hepatocytes should be included as a control for Cre-mediated effects. In Figure 2B, are those extracts derived from liver tumors or healthy tissue? At which time those tissues were collected? The authors mention they used tissues from a KO model in which HCC was induced by DEN injection but no figure showing these data is provided. I assume this refers again to their previous study, which has not been referenced in the manuscript. 

3. Figure 3D: On panel D, the authors claim that AldoB expression similarly to AKT inhibition increases the levels of p21 and p27. However, this is not clearly seen in the figure. 

4. Figure 3E-G: Aldob expression reduced both glycolysis and TCA metabolism, suggesting a general reduction in the metabolic activity of these cells. Is this caused by a reduction in glucose uptake? These data could probably be obtained from the metabolic labeling assay by monitoring the glucose consumption from the medium. 

5. Figure 3H-J: When MK2206 was administered? At the time of subcutaneous injection or once the tumors well established. This is relevant information for the therapeutic proposal. Same for SMAP in the xenograft assay shown in Figure 7.

6. Figure 4E: How Aldob enzymatic activity was determined? This is not indicated neither in the figure legend or in the methods section. 

Rev. 3:

The manuscript by He et al. investigates the adaptations in glycolytic gene expression in hepatocellular carcinoma cells. Paradoxically, they find that aldolase B is downregulated in cancer cells while glycolysis and TCA cycle is up-regulated. They point to a role for aldolase B in the control of Akt activity through PP2A dependent dephosphorylation. This novel function of aldolase B may be independent of its catalytic activity. The study contains some interesting observations. However the mechanistic insight is not fully demonstrated. The authors should consider alternative explanations and/or further validate their hypothesis. 

1- The down-regulation of aldolase B gene expression in liver cancer cells is clear. However, the authors should measure in the same cancer samples the expression of multiple aldolase isoforms, which may not be expressed in normal liver but may be reactivated during tumorigenesis. This is quite common for isoforms of many other glycolytic enzymes. Total aldolase activity should be assessed in their experimental system. 

2- All western blots should be quantified and statistically analysed.

3- The authors show in Fig. 1 an up-regulation of S6K1 phosphorylation which is at least as important as Akt. They comment that S6K1 is an indirect target of Akt activity. They should also consider the possibility that there is a general activation of both mTORC1 and mTORC2 in these cancer cells. This can be tested by looking at the phosphorylation of additional mTORC1 and mTORC2 targets. They should also measure S6K1 phosphorylation in the experimental conditions of the following figures (i.e. pharmacological inhibition of Akt). 

4- If multiple mTORC1 and mTORC2 targets are phosphorylated, the authors should consider the possibility that aldolase B may affect kinase activity in parallel of the phosphatase. 

5- In the majority of the experimental data, aldolase B expression and Akt inhibition have additive effects on the functional read-outs (metabolomics, proliferation, tumour growth: Fig. 2). The authors should consider the possibility that aldolase B and Akt may be on parallel pathways regulating tumor cell growth and metabolism. 

6- To really demonstrate the proposed mechanism, the authors should express mutants of Akt which are constitutive active and/or cannot be phosphorylated and check whether they become resistant to aldolase signalling.

---

## [Decision Letter · Decision Letter 2]

27 Oct 2020

Dear Dr Yin,

Thank you for submitting your revised Research Article entitled "Loss of hepatic aldolase B activates Akt and promotes hepatocellular carcinogenesis by destabilizing Aldob/Akt/PP2A protein complex" for publication in PLOS Biology. I have now obtained advice from the original reviewers and have discussed their comments with the Academic Editor. 

We're delighted to let you know that we're now editorially satisfied with your manuscript. However before we can formally accept your paper and consider it "in press", we also need to ensure that your article conforms to our guidelines. A member of our team will be in touch shortly with a set of requests. As we can't proceed until these requirements are met, your swift response will help prevent delays to publication. Please also make sure to address the data and other policy-related requests noted at the end of this email.

IMPORTANT:

a) Many thanks for your very thorough data provision. Please could you re-name your two supplementary data files as "S1_Data" an "S2_Data" and cite them clearly as the location of the underlying data in all of the relevant Figure legends, e.g. "The data underlying this Figure can be found in S1 Data."

b) Please insert the word "the" in the title of your article, i.e. "Loss of hepatic aldolase B activates Akt and promotes hepatocellular carcinogenesis by destabilizing the Aldob/Akt/PP2A protein complex."

- a cover letter that should detail your responses to any editorial requests, if applicable

*Copyediting*

*Published Peer Review History*

*Early Version*

Sincerely,

Roli Roberts

Senior Editor,

rroberts@plos.org,

PLOS Biology

REVIEWERS' COMMENTS:

Reviewer #1:

The addition of the reference to the paper of Li et al (2020) Nature Cancer 1:735-747 clears up the main questions raised in the initial submission as to the basis of the current study and its novelty. The authors have also added new data to satisfactorily address the other minor points that were raised.

Reviewer #2:

[identifies herself as Monica Alvarez-Fernandez]

The authors have included a set of new data in the revised version that nicely complement and further support the main findings and conclusions of the manuscript, which have also been better adjusted to the data presented in the manuscript. They have also included some additional information about methods, mouse models descriptions as well as other experimental detail, which were missed in the previous version of the manuscript. In summary, most of my concerns have been addressed, and I find the revised version to be more solid and complete. Therefore, I recommend this improved version of the manuscript for publication. 

Reviewer #3:

The authors provided further analysis that strengthen the manuscript.

---

## [Editor Report · Decision Letter 3]

13 Nov 2020

Dear Dr Yin,

On behalf of my colleagues and the Academic Editor, Marcos Malumbres, I am pleased to inform you that we will be delighted to publish your Research Article in PLOS Biology. 

PRODUCTION PROCESS

Before publication you will see the copyedited word document (within 5 business days) and a PDF proof shortly after that. The copyeditor will be in touch shortly before sending you the copyedited Word document. We will make some revisions at copyediting stage to conform to our general style, and for clarification. When you receive this version you should check and revise it very carefully, including figures, tables, references, and supporting information, because corrections at the next stage (proofs) will be strictly limited to (1) errors in author names or affiliations, (2) errors of scientific fact that would cause misunderstandings to readers, and (3) printer's (introduced) errors. Please return the copyedited file within 2 business days in order to ensure timely delivery of the PDF proof. 

If you are likely to be away when either this document or the proof is sent, please ensure we have contact information of a second person, as we will need you to respond quickly at each point. Given the disruptions resulting from the ongoing COVID-19 pandemic, there may be delays in the production process. We apologise in advance for any inconvenience caused and will do our best to minimize impact as far as possible.

EARLY VERSION

PRESS 

Kind regards,

Alice Musson

Publishing Editor, 

PLOS Biology

on behalf of

Roland Roberts,

Senior Editor

PLOS Biology